# Distributionally Robust Classification on a Data Budget

**Benjamin Feuer**                                                    *bf996@nyu.edu*
*Department of Computer Science and Engineering*
*New York University*
*370 Jay St., 11 Fl., Brooklyn, NY, 11201*

**Ameya Joshi**                                                  *ameya.joshi@nyu.edu*
*Department of Electrical and Computer Engineering*
*New York University*

**Minh Pham**                                                       *mp5847@nyu.edu*
*Department of Computer Science and Engineering*
*New York University*

**Chinmay Hegde**                                                 *chinmay.h@nyu.edu*
*Department of Computer Science and Engineering*
*New York University*

**Reviewed on OpenReview:** *https://openreview.net/forum?id=D5Z2E8CNsD*

## Abstract

Real world uses of deep learning require predictable model behavior under distribution shifts. Models such as CLIP show emergent natural distributional robustness comparable to humans, but may require hundreds of millions of training samples. Can we train robust learners in a domain where data is limited? To rigorously address this question, we introduce JANuS (Joint Annotations and Names Set), a collection of four new training datasets with images, labels, and corresponding captions, and perform a series of carefully controlled investigations of factors contributing to robustness in image classification, then compare those results to findings derived from a large-scale meta-analysis. Using this approach, we show that standard ResNet-50 trained with the cross-entropy loss on 2.4 million image samples can attain comparable robustness to a CLIP ResNet-50 trained on 400 million samples. To our knowledge, this is the first result showing (near) state-of-the-art distributional robustness on limited data budgets.

## 1 Introduction

### 1.1 Motivation

A *natural distribution shift* is defined as evaluation data which differs from the data on which a model was trained due to natural factors. Real world uses of deep image classifiers require predictable model behavior under such shifts. Unfortunately, several "standard" image classification models perform significantly worse under natural shifts (Hendrycks & Dietterich, 2019; Miller et al., 2021), in contrast with human vision (Recht et al., 2019).

Vision-Language (VL) models such as CLIP, introduced in Radford et al. (2021), have shown emergent natural distributional robustness comparable to humans across a wide range of shifts of ImageNet, at the cost of base accuracy. Jia et al. (2021) showed CLIP-like models can be carefully fine-tuned to be robust as well as achieve high base accuracy. However, such models require massive amounts of data for training, and in some cases, orders of magnitude more samples than standard supervised models (Pham et al., 2021).

These results raise challenging questions: are data scaling laws at work for robust computer vision, similar to those discovered in NLP tasks? Does robustness only emerge when models are trained on massive datasets? And is vision-language pre-training necessary for robustness? Radford et al. (2021) argue that VL pre-training in CLIP offers unique

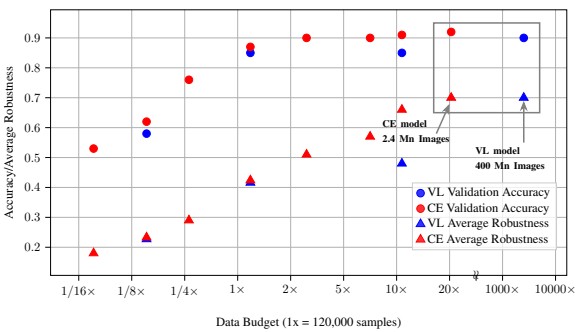 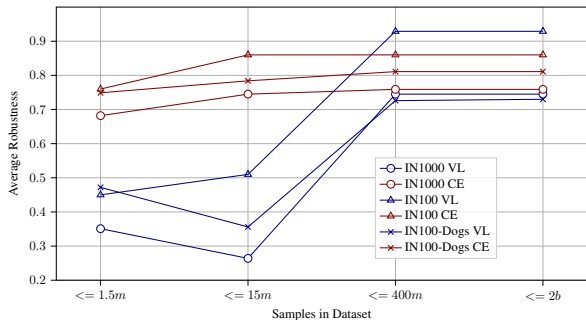

Figure 1: *(L) Under a data budget, standard CE-loss models outperform VL-loss models in both accuracy and robustness. (R) For most evaluation metrics, this effect continues in ultra-high data regimes. (L) We train ResNet-50 models using both CE-loss and VL-loss across a wide range of data scales, and find that accuracy of VL-loss and CE-loss models is extremely similar at small scales. For scaling 4X and above, CE-loss models exhibit superior robustness; the CE-loss model trained on just 2.4 Mn JANuS samples has comparable robustness, as well as comparable accuracy, to the CLIP ResNet-50 trained on 400 Mn samples. (See Tab. 1 for information on the JANuS dataset, which we create and use to train these models, and Sec. 5.1 for a detailed explanation of our methods.) (R) We compare the most robust VL-loss and CE-loss models for every tier of dataset size across three different evaluation metrics. Models trained on fewer than 1.5m samples are trained exclusively on supervised data; larger models are trained on a mix of supervised and semi-supervised data. VL-loss models are more robust on IN100. CE-loss models are more robust on IN1000 and IN100-Dogs, and when less data is available. See Sec. 5.2 for a detailed explanation of our methods. Image best viewed in color.*

advantages when compared to conventional large-data model training techniques. By contrast, Fang et al. (2022) and Nguyen et al. (2022) argue that VL robustness is a consequence of the training data diversity and quantity, with vision-language pretraining playing little role.

In most real-world applications, data is limited, and unlikely to be accompanied by informative natural language captions. For example, the PCam medical imaging dataset from Veeling et al. (2019) has only around $320,000$ images. Nevertheless, distributional robustness is of paramount importance in the setting. What can be done to train robust models in data-limited settings without access to informative captions? Can we leverage other attributes of model training which have largely been disregarded in the distributional robustness literature, such as architecture, model size, and image resolution?

## 1.2 Our Contributions

Our objective in this paper is to clearly delineate the potential factors influencing distributional robustness for image classification. To achieve this, we evaluate a vast suite of existing models trained on diverse data budgets, supplementing with dozens of new models trained from scratch. Overall, our key contributions are as follows:

1. We introduce JANuS (Joint Annotations and Names Set), a new class-balanced dataset with images, labels, and captions. To our knowledge, this is the first such dataset of its kind. (See Tab. 1).
2. We conduct ablation studies of several categories of image classification models using JANuS as a controllable training dataset. For the first time, we show that it *is* possible to train highly robust and accurate models, using conventional cross-entropy (CE) loss, even when both data and model size are limited (See Fig. 1).
3. We conduct the largest meta-analysis (to date) of robustness of image classification models (numbering over 650), including many recent architectures, and show that even with relatively modest model and data scaling (compared to Brown et al. (2020), Radford et al. (2021)), one can achieve robust classification performance. (See Fig. 2 and Fig. 3.)
4. We outline useful heuristics to improve distributional robustness when data budgets are limited. (See Sec. 7).
5. In order to enable future research and reproducibility, we release our code, our dataset, and a complete enumeration of our results for all models in the study (see supplemental attachments).

## 2 Related Work

Our paper follows a series of recent works studying robustness under distribution shift in the context of image classification (Recht et al., 2019; Taori et al., 2020; Miller et al., 2021; Fang et al., 2022; Nguyen et al., 2022). This line of inquiry into distributional robustness focused on the linear fit between in-distribution and out-of-distribution accuracy found between common image classification datasets (such as ImageNet) and their distribution shifts. In contrast to most of these earlier papers, our analysis takes place in a realistic setting where models are trained on a wide range of datasets. Therefore, following the results in Nguyen et al. (2022), we do not use linear fit measures in our analysis, instead relying on average out-of-distribution robustness.

Jia et al. (2021); Pham et al. (2021) showed that human-level distributional robustness is possible even as base accuracy approaches state-of-the-art, as long as sufficient data is available for training. The gains are not limited to CLIP; other VL-loss (vision-language loss) functions also achieve strong distributional robustness (Yu et al., 2022; Wang et al., 2022b). We discuss some of these alternate approaches in Appendix Sec. A, while noting that none of these exhibit superior robustness than CLIP.

The internals of CLIP differ from those of typical models in several important ways: the choice of loss function, the training dataset, and the use of natural language captions as labels. However, identifying which of these differences lead to CLIP's outstanding robustness is still an open question. Recent works have addressed this question from different angles. Fang et al. (2022) argue that intrinsic diversity of training image data is the main source of the distributional robustness gains of VL models in the zero-shot setting, with factors such as language supervision contributing little to no distributional robustness. However, in a different (transfer learning) setting, Santurkar et al. (2022) argue that, given a sufficiently large pretraining dataset and descriptive, low variability captions, contrastively trained VL models are more robust than self-supervised image-only models trained with the SIMCLR-loss. We conduct controlled comparisons between vision-language classifiers and conventional classifiers, and find that when controlling for data quantity and diversity, high accuracy VL-loss models are actually *less* robust than high accuracy CE-loss models.

Nguyen et al. (2022) is an important precursor to our work. Their extensive experiments on vision-language models in the low accuracy regime showed that controlling for the pretraining dataset was essential for understanding distributional robustness. We extend this understanding, and show that model architecture, size, image resolution, and even the label set selected for the classification problem can all have substantial effects on robustness. Finally, unlike Nguyen et al. (2022), all our results are shown in both low and high accuracy regimes, and across different test sets.

In their paper investigating the role of language on robustness, Fang et al. (2022) introduced ImageNet-Captions, which added Flickr-captions to nearly 450,000 ImageNet images. We extend this work by introducing JANuS, which add over 50,000 new human-supervised samples to 100 classes in ImageNet-Captions in order to rebalance the classes, as it has been shown that CE-loss models often struggle with imbalanced classes (Phan & Yamamoto, 2020).

## 3 Preliminaries

**Metrics for distributional robustness.** Our primary metric is *average robustness* (abbv: Avg. Rob.), which is the average test-set accuracy of a model on a set of distribution shifts. Although this measure is easy to interpret, it can conceal substantial performance differences between shifts. Another metric we use is *effective robustness*, introduced by Taori et al. (2020), primarily to situate our work within the existing literature. This metric is a graphical tool to describe how robust a model is on natural distribution shifts. For human vision, a graph of base-versus-shift test accuracy follows the $y = x$ trendline; for neural networks, this trend-line generally is parallel to but below $y = x$. Finally, we include *Effective Robustness Ratio* (abbv: E.R.R.), from Feuer et al. (2022) in our appendix tables. This is defined as the ratio of average robustness over base task accuracy. This is an effective measure for comparisons among models with roughly similar base accuracy.

**Fine-tuning.** Fine-tuning is the extremely common practice of initializing the weights of a model to values attained during pretraining, and then adjusting them based on a new dataset. It has been shown that large-scale pretraining can dramatically improve the base accuracy of computer vision models when compared to random initialization (Dosovitskiy et al., 2021; Steiner et al., 2022), and that the ImageNet dataset is very well suited to this task (Kornblith et al., 2019). However, distributional robustness generally does not improve in proportion to the gains in base accuracy. (Recht et al., 2019) In fact, Radford et al. (2021) found that pretraining and fine-tuning rapidly can *erode*

| Dataset | G.T. Label | Machine Label | Caption Source | Supervised | Filtered | Balanced |
|---|---|---|---|---|---|---|
| ImageNet-100 (IN100) | ✓ | ✓ | Flickr, BLIP | ✓ | Human | ✓ |
| OpenImages-100 (OI100) | ✓ | ✓ | Flickr, BLIP, annotated | ✓ | None | X |
| LAION-100 (LAION100) | X | ✓ | alt-text | X | CLIP | X |
| YFCC-100 (YFCC100) | X | ✓ | Flickr | X | Algo | X |

Table 1: **_The JANuS dataset allows for controlled comparisons between models in a high accuracy regime._** *The experiments in Fig. 1 (L) were conducted using a combination of the four main datasets in JANuS, which are described here.* **G.T. Lbl.** *indicates the presence of human-annotated ground truth labels in the dataset.* **Machine Lbl.** *indicates availability of synthetic labels; labeling strategies are detailed in 4.3.* **Caption Src.** *lists the sources for captions in the dataset.* **Supervised** *indicates when ground truth labels exist for the dataset. CE-loss models benefit most from supervised data.* **Filtered** *indicates when the dataset contents were processed in some way prior to inclusion. VL-loss models struggle on unfiltered data.* **Balanced** *indicates whether the dataset is approximately class-balanced.*

distributional robustness, even as base accuracy increases. Zhai et al. (2021); Wortsman et al. (2022a;b) closed the gap but were unable to reproduce the zero-shot robustness attained by Radford et al. (2021). Given the limited efficacy of fine-tuning, the majority of our results are reported on models trained from scratch. We confine our specific remarks on transfer learning to Sec. A.

**Glossary.** For ease of understanding, we provide a glossary of common terms and abbreviations.

*Loss functions.* We examine models trained with two types of losses. *VL-loss* refers to the InfoNCE loss used by CLIP (Radford et al., 2021). *CE-loss* is the typical cross-entropy loss used to train image classification models.

*Label types.* CE-loss models use *integer* labels (referring to discretely labelled classes), and VL-loss models use *caption* labels. We refer to human-annotated labels (whenever available) as *ground-truth*. We refer to labels generated by automated processes as either *synthetic* or *subset-matched* (defined below in Sec. 4).

*Data filtration.* We define *data filtration* as any strategy which sub-selects image-caption pairs.

# 4 JANuS: A New Benchmark Dataset for Robust Model Training

Training large-scale image classification models from scratch on existing benchmarks may present resource challenges for academic researchers, particularly vision-language models with dual architectures. Prior papers such as Fang et al. (2022) resolve this by obtaining low- or medium-accuracy results, and using the linear fit hypothesis of Miller et al. (2021) to project those results to high accuracy regimes. However, as observed in Nguyen et al. (2022), trends observed in low-accuracy regimes may not persist in high-accuracy regimes, unless the training dataset, loss function, and label set are controlled for across different models.

For this reason, we postulate that a 100-class, broad scope problem is ideal for comparative studies of robustness. However, no training dataset exists which is designed for 100-class ImageNet problems and is sufficiently diverse to train high accuracy, high-robustness models with both VL-loss and CE-loss objectives.

To resolve this, we introduce JANuS (Joint Annotations and Names Set), a collection of four new training datasets with images, labels, and corresponding captions. Each dataset in JANuS builds upon an existing dataset by selecting or adding data from a known data source. Data sources for which ground truth labels exist are filtered by class. For data sources where ground truth labels do not exist, we use a technique called *subset matching* to prefilter JANuS; a detailed explanation of this technique can be found in Sec. H. The constituent datasets are the following:

1. **ImageNet-100 (IN100):** The 100 largest ImageNet-Captions classes from Fang et al. (2022), followed by class rebalancing by addition of over 50,000 new image samples annotated with human-authored ground-truth labels.
2. **OpenImages-100 (OI100):** A subset of the OpenImages dataset, Kuznetsova et al. (2018), with restored original Flickr-captions, and new BLIP-captions; samples selected by mapping human-labeled OpenImages-100 classnames to ImageNet-100 classnames.

3. **LAION-100 (LAION100):** A subset of the unlabeled LAION dataset, Schuhmann et al. (2021a), with samples selected via subset matching on ImageNet-100 classes.
4. **YFCC-100 (YFCC100):** A subset of the unlabeled YFCC dataset, Thomee et al. (2016), with samples selected via subset matching on ImageNet-100 classes.

We compare some of the key properties of each component of JANuS in Tab. 1.

The most important new contribution in JANuS is ImageNet-100, which is, to the best of our knowledge, the only version of ImageNet which duplicates the original distribution's class balance and supervision properties (ImageNet is not perfectly class balanced, but it does not contain any long-tail classes; all classes in ImageNet have at least 750 samples), while also being fully captioned with original web-scraped labels.

Every JANuS sample has descriptive captions *as well as* class labels (either as human annotated or synthetic labels).

Furthermore, either VL-loss or CE-loss models trained on JANuS alone can achieve high validation accuracy, making it possible for the first time to compare model distributional robustness while controlling for base accuracy.

These properties enable JANuS to be used to fairly compare both image-only and image-text training approaches while controlling for dataset size and quality, making it a useful new benchmark for robustness comparisons.

### 4.1   IN100 performance is comparable to ImageNet.

In order to ensure that the baseline performance of VL-loss and CE-loss models is comparable on IN100 and the standard ImageNet despite the 50,000 newly labeled images, we train a VL (using the standard "A photo of a $CLASS-NAME" prompt) and CE-loss model from scratch on IN100, and compare it to a CE-loss model trained for 256 epochs on the same 100-class subset of ImageNet. Controlling for size, we find that our dataset performs slightly worse than the baseline, but considerably better than that subset of ImageNet-Captions (Fang et al., 2022) alone.

### 4.2   Dataset Construction

The 100 classes in JANuS were selected randomly from a subset of all classes with more than 600 captions available in ImageNet-Captions (Fang et al., 2022). The list of classes selected is available in Sec. J. We acknowledge that as with any filtration strategy, this class selection approach favoring classes with more captioned images may introduce a potential selection bias (which might spuriously correlate with accuracy or robustness). However, we feel that the risk of this is outweighed by the many benefits of having such a dataset publicly available for future studies.

### 4.3   Supervision strategy

In Table 8 in the Appendix, we discuss in detail the supervision (labeling) strategy used for JANuS, with a per-class breakdown of each class. An overview of the supervision process is as follows:

- All image samples were supervised by the authors of this paper.
- Samples were sourced from Flickr using the available API, sorted by 'interesting', with safesearch enabled, searching only samples with Creative Commons licenses.
- Additional filtering terms were passed to the API in order to eliminate common confounds in the search terms.
- After the search term was selected, items were downloaded in bulk.
- All downloaded samples were then individually tagged by the researchers as either "IN-class" or "out-of-class", using reference photographs from each class as a baseline comparison.

We found that classes varied in several respects:

- Some classes had far greater availability than others (ranging from 450,000 to 283 available samples).
- Some classes were much cleaner than others (ranging from 100 percent clean to around 25 percent).
- Some classes tended to be the 'subject' of images (such as dog breed) while others, such as mashed potato, tended to be featured as secondary items in the 'background' of a image saple.

# 5    Experimental Design

Our motivating goal in this paper is to identify which factors are most decisive in determining distributional robustness in data-limited regimes. We enumerate several possibilities below:

**(Q1)**  Are models trained on **more samples** more robust than models trained on **fewer samples**?
**(Q2)**  Are models with **more parameters** more robust than models with **few parameters**?
**(Q3)**  Are **VL-loss** models more robust than **CE-loss** models?
**(Q4)**  Are **ViT** models more robust than **convolution-based** models?

We address each of this questions by designing careful experiments to evaluate model robustness. For each question, we conduct two distinct types of evaluations. Approach 1 is a small-scale evaluation of models trained under highly controlled settings. Approach 2 is a large-scale meta-analysis of publicly available pretrained models. We note that both approaches have strengths and limitations, but we believe that trends found to be consistent in both approaches are likely indicative of future model performance. Where trends differ, as in Q4, we try to understand the root cause of these differences.

## 5.1    Approach 1: Controlled comparisons on JANuS-trained models

In our first set of experiments, we measure accuracy and robustness of various models trained on our proposed JaNUS dataset. Using this approach, we can control for confounding factors by changing one aspect of the training regimen at a time, and we can control for the pretraining dataset. This is important if we are to truly isolate architectural/algorithmic factors influencing robustness, which has thus far been absent from many existing studies of distributional robustness in the literature[1].

In all our model training experiments, we train with mixed precision, at a batch size of 256, and do not use gradient clipping. We use the AMP library to implement the training process. Model hyperparameters are chosen via grid search. Models are typically distributed across a single node with 4 NVIDIA A100 GPUs; our largest models were trained on 16 NVIDIA GPUs. All JANuS models were trained for 256 epochs. Following Santurkar et al. (2022), we use SimCLR augmentations (resize, crop, flip, jitter, blur, grayscale) rather than CLIP augmentations (resize and crop) for model training. Our code is publicly available for reproducibility purposes.

**Models.**   We train over twenty-five models on the JANuS dataset; the complete list can be found in Tab. 9. Our baseline model against which all variations are compared is a ResNet-50 with a 1000-class linear classification head (He et al., 2015). Specific variations are discussed in more detail below.

**Labels.** We evaluate on a single, broad-scope label set of 100 classes corresponding to ImageNet-100 (IN100), which is the first constituent dataset of JANuS; see Sec. 4 above. We refer to the validation set for IN100 as IN100-Val.

**Shifts.** Following Radford et al. (2021), we focus on the following four distribution shifts: Imagenet-Sketch, Imagenet-R, Imagenet-A, and Imagenet-V2, for our evaluation. For ImageNet-R and ImageNet-A, which are subsets of ImageNet, we evaluate only the 35 shared classes. We reference the validation sets for these shifts on IN100 as IN100-V2, IN100-S, IN100-R, and IN100-A. Additional details on the datasets, distribution shifts, and class indices are in Appendices Sec. C, Sec. D, and Sec. J.

**Metrics.**   We evaluate our models every 32 epochs, and report the "best" average robustness across all shifts, with "best" being determined by the model's peak performance across all evaluated checkpoints.

With these basics in place, we design experiments specific to questions **(Q1)-(Q4)**.

**(Q1) Comparing across dataset size.** We report dataset size in approximate multiples of the size of JaNUS's IN-100 training set. For example, a 1x JANuS model is trained on approximately $120,000$ samples. A 10x JANuS model is trained on approximately $1,200,000$ samples. We primarily compare models at 1x, 2x, and 10x scales. In addition to directly reporting the effects of scaling dataset size, we report the secondary effects of scaling dataset size when changing parameter count, loss function and architecture.

---

[1]Controlling for a fixed training dataset has some limitations. We cannot evaluate the effects of scaling pretraining data beyond the training set size of JaNuS (2.4 million images); we cannot evaluate the effects of pretraining on a wider range of label classes; and we cannot evaluate models trained on classes outside ImageNet-100. We address these in the larger meta-analysis presented in Sec. 5.2.

**(Q2) Comparing across parameter counts.** Our models comparing the effects of scaling parameter count in CE-loss models are a ResNet-26, a ResNet-50, and a ResNet-50x4 model with identical 1000-class linear heads. The latter is a ResNet-50, scaled up 4x according to the EfficientNet scaling rule (Radford et al., 2021; Santurkar et al., 2022).

**(Q3) Comparing VL and CE-loss.** Our models comparing CE-loss and VL-loss are a ResNet-50, and a CLIP architecture with a ResNet-50 vision backbone. The only difference in the two architectures is that for CE-loss models, we append a ResNet-50 with a 1000-class linear head, whereas our VL-loss model uses a text transformer for classification. Because of this, our vision-language models have significantly higher parameter counts than standard computer vision models; we allow for this difference when controlling for parameter count – EG, we compare a RN50 using VL-loss to a RN50 using CE-loss, rather than comparing the RN50 using VL-loss to a parameter-equivalent RN50x4 using CE-loss. We choose this approach because, as noted in Radford et al. (2021); Santurkar et al. (2022), the difference does not measurably affect distributional robustness.

**(Q4) Comparing ViTs and convolutional models**. When comparing ViTs to convolutional models, we compare the ViT-S-16 model introduced in Dosovitskiy et al. (2021) to our ResNet-50 baseline. ViT-S-16 has approximately 88% as many parameters as the ResNet-50. We do not ablate the effects of this difference on architecture performance, but our scaling experiments across parameter counts lead us to believe that the effect is minor, compared to other factors.

### 5.2 Approach 2: Large-scale Meta-Analysis of Pre-trained Models

Approach 2 is a large-scale meta-analysis of publicly available pre-trained models, similar to the setting previously adopted in Taori et al. (2020); Miller et al. (2021). This approach illuminates the effects of scaling pretraining data across large ranges, as well as the effects of pre-training on a wide range of classes. While this enables us to compare trends across hundreds of models, we note that such observational studies cannot carefully control for architectural and/or algorithmic confounding factors. Our complete results can be found in our main results table in the supplementary attachment, 1_captionnet_in1k_model_results_and_metadata. Our experiments are designed as follows.

**Models.** We compare over 650 models drawn from the pytorch-image-models (timm) repository (Wightman, 2019), the open-clip repository (Ilharco et al., 2021), and a few newly trained models[2]. Architectural details about the models included in this approach are listed in Sec. G. We conduct no additional fine-tuning of these models.

**Labels.** We report evaluation results on the label set of ImageNet-1K, which is the standard benchmark in the majority of existing works. We refer to this validation set as IN1000-Val. In addition, we also report results on the label set of ImageNet-100 (defined above), which is a broad scope label set; we also evaluate on a fine-grained subset of ImageNet-1K that we call *ImageNet-Dogs* (IN100-Dogs, for short). This consists of a subset of ImageNet class labels corresponding to 100 dog breeds, and we do this in order to provide an alternative view of 100-class classification using ImageNet models. We refer to this validation set as IN100-Dogs-Val.

**Shifts.** We utilize the same four shifts as in Approach 1. We abbreviate these shifts following the same pattern as described in Approach 1; IN1000-V2, IN100-Dogs-V2, IN1000-S, IN100-Dogs-S, etc.

**Metrics.** The metrics are identical to those used in Approach 1, except that we provide the average for each label set individually, rather than reporting the aggregate over all label sets. We do not report the average of all models fitting a certain category, as that would skew the evaluation against older releases in the repository. Instead, we report the average of the *ten highest performing* models in each category. For example, if we are evaluating the effects of parameter count, we first bin the models by parameter count, and then report the average of the ten best performing models in each bin. We also present individual model scores in scatterplot format in our figures.

## 6 Results, Trends, and Ablations

We now present a series of experimental results that address each of the questions **Q1-Q4** listed in Sec. 5. For each question, we present our key findings, and then support these from our experimental results with Approach 1 (controlled experiments on JANuS) and Approach 2 (meta-analysis with pre-trained models.)

---

[2]We intended to restrict our meta-analysis to models available in existing public repositories; however, there seems to be a dearth of models trained with a number of samples between 15M and 300M. We fill this gap by training several models on increasing-size subsets of the LAION-2B dataset.

| Avg. robustness by label set | | | | |
|---|---|---|---|---|
| Model | IN1000 Val. | IN1000 Avg. Rob. | IN100 Avg. Rob. | IN100-Dogs Avg. Rob. |
| **VOLO-D5-224 (1.2 Mn)** | .857 | .594 | .725 | .723 |
| VGG-16 (1.2 Mn) | .716 | .266 | .402 | .433 |
| Avg. robustness by shift | | | | |
| Model | V2 | S | R | A |
| **VOLO-D5-224 (1.2 Mn)** | .814 | .55 | .652 | .707 |
| VGG-16 (1.2 Mn) | .66 | .251 | .363 | .195 |

Table 2: ***Choice of architecture can strongly impact model performance in controlled-data settings.*** *In contrast to recent claims in the robustness literature, we find that factors other than data can strongly affect distributional robustness. Specifically, when we compare a VGG-16 (Simonyan & Zisserman, 2014) baseline to a recent VOLO model (Yuan et al., 2021), we see substantial gains in robustness. The gain is not localized to particular shifts, nor does it scale in proportion with base accuracy.*

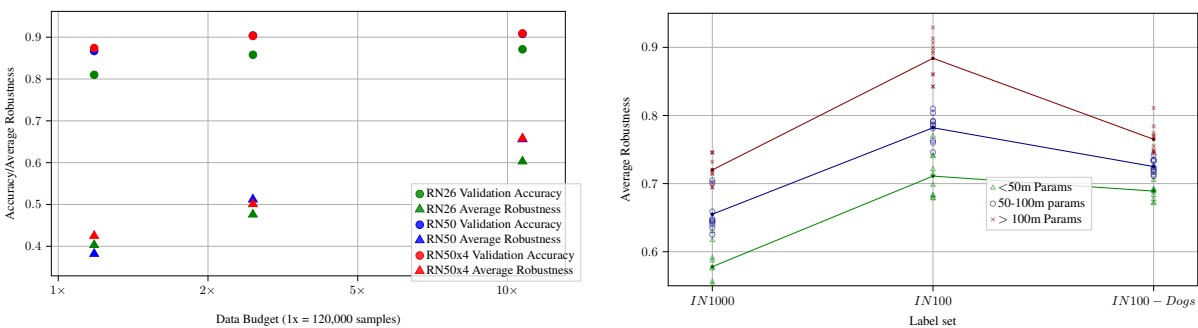

Figure 2: *(L) Effects of scaling parameter count (Approach 1). (R) Effects of scaling parameter count (Approach 2). (L) We find that increasing the parameter count has a positive effect on average robustness, but the effect is limited on small data budgets, and can invert when data is very limited. (R) When we compare the average robustness of models in the study by their parameter count using approach 2, we see reliable improvements as model size increases; larger models are more robust on all label sets. Image best viewed in color.*

**(Q1) Are models trained on more samples more robust than models trained on fewer samples?**

Increasing quantity of training data is known to positively impact robustness. Fang et al. (2022) argued that diverse, and presumably large, training distributions nearly exclusively account for the strong robustness of VL-loss models, and Taori et al. (2020); Miller et al. (2021) contend that among CE-loss models, factors other than data have little impact on robustness, except insofar as they increase base accuracy. If so, then we should expect that data (and data alone) can impact distributional robustness.

The findings in Fang et al. (2022) relied on prior work from Taori et al. (2020) in order to project their findings from low to high accuracy models. However, the work of Taori et al. (2020) predates the introduction of some unusually robust architectures, including Yuan et al. (2021); Hatamizadeh et al. (2023); Liu et al. (2022). The linear trend in the accuracy/robustness space described in that work no longer holds for all shifts and all models, even when controlling for pretraining dataset; we document this in (Q4), as well as in Appendix Tab. 10, Tab. 13, Tab. 11.

*Overall, we find that although it is not the only factor, dataset size is still an important determinant of distributional robustness.*

**Evidence using Approach 1.** We find that increasing the quantity of training data has a reliable and positive effect on average robustness. Specifically, our baseline model at 1x scaling averages **.42** robustness, whereas with 2x scaling, it

improves to **.51**. The marginal improvements decline at 10x (**.66**) and 20x (**.7**) scales, suggesting the returns of data scaling diminish at the extremes. See Table 9 for additional results in this vein.

**Evidence using Approach 2.** This finding persists when we make large-scale comparisons of over 650 pre-trained models. For example, a standard ResNet-50 model trained on ImageNet-1K (i.e., a 10x data budget in our terminology) achieves a robustness of **0.504**. The SWSL ResNet-50 model trained by Mahajan et al. (2018) on a 7800x data budget achieves robustness of **.74**. The full table is too large to parse; please refer to the supplementary attachment, 1_captionnet_in1k_model_results_and_metadata.

### (Q2) Are models with more parameters more robust than models with fewer parameters?

Earlier works such as Mahajan et al. (2018); Radford et al. (2021); Tan & Le (2019) have shown empirically that parameter count, in conjunction with other factors, can affect both validation accuracy and distributional robustness. However, Miller et al. (2021); Taori et al. (2020) found that when pretraining on ImageNet, increasing parameter counts improved robustness only within certain predictable limits.

**Evidence using Approach 1.** Our findings here agree with both groups; on JANuS, we find that increasing the parameter count has a positive effect on average robustness, but the effect is limited on small data budgets; when dataset size is very limited (1x or below), increasing parameter count can actually decrease robustness. (see Fig. 2)

**Evidence using Approach 2.** In our analysis, 438 of the models have fewer than 50 million parameters, 126 have between 50 and 100 million, and 86 have over 100 million parameters. Of the top 100 most robust models, 13 have fewer than 50 million, 33 have between 50 and 100 million, and 54 have over 100 million parameters.

In Fig. 2 (L), we compare model performance on our three evaluation metrics, grouped by parameter count. We find that average robustness improves reliably with model size across all evaluations. The gains are most significant on IN100.

### (Q3) Are VL-loss models more robust than CE-loss models?

Large VL-loss models such as those of Radford et al. (2021); Pham et al. (2021) have been conventionally presented as robust generalist models which can handle arbitrary (open vocabulary) classification tasks. However, to our knowledge no large-scale validation studies quantifying the value of VL-training in terms of robustness have been reported thus far in the literature. We fill this gap.

*Overall, when controlled for other factors, we find that VL-loss models are no more robust than CE-loss models.*

**Evidence using Approach 1.** We train VL-loss and CE-loss models from scratch on JANuS and evaluate them on IN100 across a wide range of data scales.

Ground-truth labels have been shown to improve base accuracy of VL-loss models. Fang et al. (2022) found that a ResNet-50 VL-loss model trained on ImageNet-1k with ground truth labels ("A photo of the CLASSNAME") achieved accuracy and robustness parity with a CE-loss ResNet-50 for IN1000 classification. In Fig. 1, we show that this is also the case for ResNet-50 models trained and evaluated on IN100.

However, we see that at no point does VL-loss offer a robustness advantage. On the contrary, at larger data scales, VL-loss is at a disadvantage; a CE-loss model trained on just 2.4M JANuS samples has comparable robustness, as well as comparable accuracy, to CLIP ResNet-50 trained on 400M samples.

**Evidence using Approach 2.** For dataset sizes below 400M samples, we find no reliable evidence that VL-loss models are more robust than CE-loss models in absolute terms on IN1000 or IN100-Dogs; see Fig. 2 (R). We also note that VL-loss models have lower base accuracy on these problems. VL-loss models do show a robustness advantage on IN100.[3]

---

[3]We postulate that this discrepancy may be because smaller label sets are easier to disambiguate using natural language, and provide two experiments as evidence. First, in Sec. F, we provide per-class accuracies for a VL-loss and CE-loss ResNet-50 trained on many samples, and note that several of the classes on IN1000 where VL-loss models substantially underperform CE-loss models have identical natural language descriptions; in IN1000, OpenAI's classnames include two classes labeled "missile" and two classes labeled "sunglasses", reflecting ambiguities in the underlying problem (Radford et al., 2021; Beyer et al., 2020). Second, we train a VL-loss model on 10 million YFCC samples, filtering out all samples whose caption contains a matching term with an ImageNet class; the resulting model has, in theory, not been trained on any of the classes on which we

Above 400M samples, our investigation is limited by the fact that relatively few public models have been trained on such huge datasets; our largest CE-loss models were trained on half the data of the largest VL-loss models, and they have few other architectural features in common. The limited evidence we have indicates that VL-loss models might enjoy a robustness advantage at very high data regimes (See Fig. 1); however, this might be confounded with the fact that no publicly released ViTs have been trained with CE-loss at this scale.

| Model | Dataset | IN1000 Val. Acc. | IN100 Val. Acc. | IN100-Dogs Val. Acc. |
|---|---|---|---|---|
| ResNet-50 | YFCC-10Mn-N.I. | .127 | .329 | .034 |
| ResNet-50 | YFCC-15Mn | .324 | .741 | .086 |

Table 3: **VL-loss models trained on web-scraped caption labels learn classes unevenly.** *VL-loss models learn a lot about broad distinctions between classes from captions, and little about fine-grained class boundaries. This finding holds even when we remove all samples which match with any term in the OpenAI ImageNet classnames from the YFCC-15Mn dataset (YFCC-10Mn-N.I.). Robustness scores can be found in our main results table in the supplementary attachment, 1_captionnet_in1k_model_results_and_metadata.*

### (Q4) Are ViT models more robust than convolution-based architectures?

In Tab. 2, we demonstrate that the marginal robustness gain of going from a VGG-16 from Simonyan & Zisserman (2014) to VOLO-D5-224, the best model using Approach 2, trained on ImageNet-1k alone, at 224px image resolution, is much greater than the gains of data scaling alone. This finding motivates a consideration of which architectures are likely to be more robust, given a data budget. One natural way to divide approaches is to compare models based on the vision transformer architecture to convolution-based approaches.

*Overall, we find that architecture can strongly impact distributional robustness, controlling for all other factors.*

**Evidence using Approach 1.** We observe that a ViT-S-16 performs worse than the parameter-equivalent RN50 on JANuS. This finding is consistent from the smallest to the largest scales we consider. (see Fig. 3) This finding is in keeping with Dosovitskiy et al. (2021), who found that ViTs only reach parity with CNNs after very large scale pretraining. Unlike Dosovitskiy et al. (2021), we find that pretraining and fine-tuning does not necessarily help close the gap. ViTs are more robust and more accurate than ResNets after pre-training but before fine-tuning. After fine tuning, the advantage disappears. (see Tab. 4)

| Dataset | Data Scale | Fine Tune | RN50 IN100 Val. Acc. | RN50 IN100 Avg. Rob. | ViT-S-16 IN100 Val. Acc. | VIT-S-16 IN100 Avg. Rob. |
|---|---|---|---|---|---|---|
| IN100 (JANuS) | 1x | FALSE | 0.87 | 0.424 | 0.713 | 0.256 |
| IN100+OI100 (JANuS) | 2x | FALSE | 0.904 | 0.511 | 0.785 | 0.325 |
| IN100+LAION100 (JANuS) | 4x | FALSE | 0.901 | 0.57 | 0.787 | 0.376 |
| IN1k | 10x | FALSE | 0.956 | 0.505 | 0.942 | 0.474 |
| IN1k+IN100 | 10x | TRUE | 0.937 | 0.564 | 0.926 | 0.502 |
| JANuS | 10x | FALSE | 0.908 | 0.655 | 0.823 | 0.464 |
| JANuS+YFCC | 20x | FALSE | 0.927 | 0.702 | 0.878 | 0.576 |
| IN21k+IN1k | 100x | FALSE | 0.939 | 0.538 | 0.957 | 0.543 |
| IN21k+IN1k+IN100 | 100x | TRUE | 0.924 | 0.499 | 0.926 | 0.501 |

Table 4: **ViTs are usually less accurate and robust than ResNets on data budgets.** *ViTs consistently underperform ResNets across a wide range of data budgets. After pretraining but before fine-tuning, ViTs are more robust and more accurate than ResNets. After fine tuning, the advantage disappears. This indicates that the robustness advantage enjoyed by massively pretrained ViTs may not be preserved during fine-tuning.*

**Evidence using Approach 2.** The 650 models in our analysis include 385 convolution-based architectures and 204 vision transformers; despite the relative overrepresentation of convolution architectures in the study, of the 100 timm

---

evaluate it. When we do this, we find that the resulting model achieves just 3% accuracy on IN100-Dogs, but achieves 33% accuracy on IN100; in the absence of ground truth labels, the model 'guesses better', in essence, when classes are dissimilar. See Tab. 3.

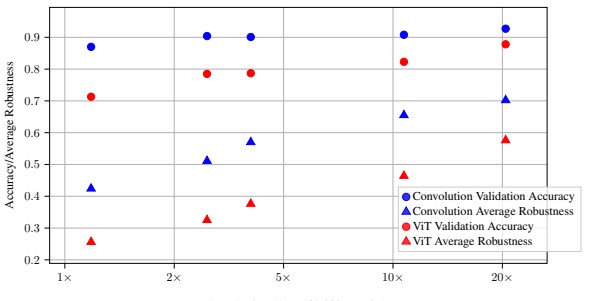 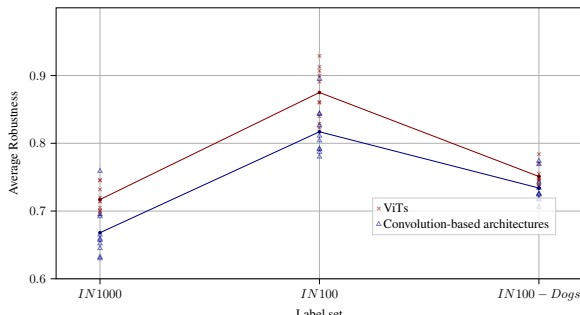

Figure 3: *(L) Effects of architecture (Approach 1). (R) Effects of architecture (Approach 2). (L) We compare average robustness of a ViT-S-16 and a ResNet-50 on JANuS, using a range of data scales, and find that the ViT underperforms the ResNet throughout. (R) We evaluate average robustness of models in the study when grouped by architecture, and find ViTs outperform convolution-based architectures. Individual marks on the graph represent the average robustness of the ten most robust convolution-based and transformer-based models, respectively. Trend lines follow the group average. Image best viewed in color.*

models with the highest average robustness on IN1000, 60 are ViTs and 40 are convolution-based architectures. On IN100, the split is 70 / 30, and on IN100-Dogs, 72 / 28.

Comparing the top 100 most robust models for each problem, we find that ViTs are, on average, substantially more robust, and the advantage grows at massive data scales (see Fig. 3).

*Interpreting the trends.* Aside from the aforementioned limitations of ViTs when training on limited data, we attribute the strong performance of ViTs in Approach 2 to the recent emergence of new variations on the ViT architecture which are more robust than the standard ViTs. Recent effective interventions have modified the attention mechanism Yuan et al. (2021); Hatamizadeh et al. (2023) or the pretraining strategy Bao et al. (2022); Touvron et al. (2022) to achieve substantial gains in this regard. See Tab. 5 for further details.

| Model | IN100-Val | IN100-V2 | IN100-S | IN100-R | IN100-A | IN100 Avg. Rob. |
|---|---|---|---|---|---|---|
| in100-ViTS16 | 0.713 | 0.584 | 0.124 | 0.204 | 0.11 | 0.256 |
| in100-DeITS16 | 0.756 | 0.643 | 0.164 | 0.222 | 0.137 | 0.292 |
| in100-GCViTS16 | 0.803 | 0.702 | 0.378 | 0.356 | 0.143 | 0.395 |

Table 5: *Recent ViT-based architectures are more robust and accurate on data budgets. Although they do not match the performance of ResNets, recent improvements to the ViT such as DeIT from Touvron et al. (2022) and the GC-ViT from Hatamizadeh et al. (2023) improve dramatically on standard ViTs. All models listed here were trained on IN100, at a 1x data budget, for 256 epochs.*

# 7 Discussion and Useful Heuristics

## 7.1 Summary of Key Findings

Our detailed experimental results in Section 6 demonstrate the effects of various architectural, algorithmic, and data-dependent factors on distributional robustness of image classification models. We conclude with a summary of key takeaway points, along with a list of suggested useful heuristics for training robust models in future applications.

1. Motivated by our findings for Q1, *scaling data quantity* is likely to have the most reliable impact on distributional robustness during training.
2. Motivated by our findings for Q3 (Approach 2), for *few-class problems* where either the classes themselves or their visual properties (color, shape, etc.) are easily disambiguated using text alone, we conclude that the most robust and most efficient approach is to *use a zero-shot VL model.*

3. Motivated by our findings Q2, Q3 and Q4, for *fine-grained classification problems*, for problems with ambiguous class names and many-class problems, the best approach when *training from scratch* is a *CE-loss model* with a *tailored convolution-based architecture* such as ConvNeXt, scaled to parameter counts at which returns diminish.

## 7.2 Future Work

As computer vision models and datasets grow in size, and multimodal generative models such as OFA from Wang et al. (2022a) introduce and solve new, complex problems, the task of developing a prescriptive set of "scaling laws" for emergent distributional robustness will only increase in importance (Cherti et al., 2022). Equally important will be comparing the behavior of models on distribution shifts for datasets other than ImageNet. Finally, a comprehensive understanding of model performance on more challenging, long-tailed classification problems (such as iNaturalist) will shed more light on the robustness profile of models in the real world.

## Acknowledgements

This work was supported in part by the National Science Foundation (NSF) under grant CCF-2005804, an AI Research Institutes grant supported by NSF and USDA-NIFA under the AI Institute for Resilient Agriculture grant 2021-67021-35329, and the Department of Energy under grant DE-EE0009105.

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

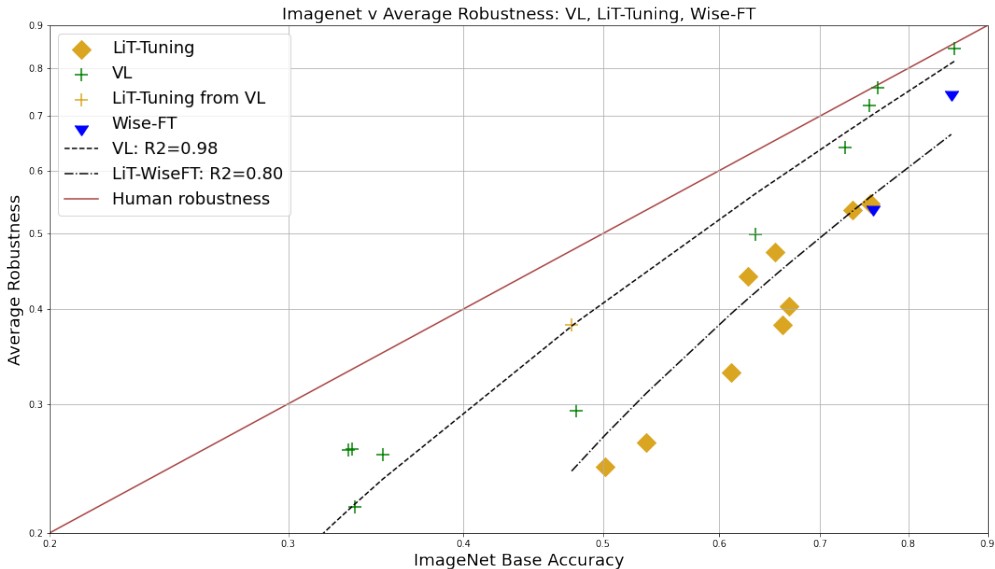

Figure 4: **Wise-FT, optimized to balance id/ood accuracy, fits the LiT-tuned effective robustness line.** Both Wise-FT and LiT-tuning exhibit lower effective robustness than conventional vision-language pretraining.

## A  Transfer learning in vision-language models

Another approach to robust classification in VL is using some form of transfer learning instead of training from scratch. The robustness advantages of transfer learning are well understood in conventional computer vision (see Kolesnikov et al. (2019)), and many recent model releases include variants which are pretrained on ImageNet-21k. Such models generally exhibit improved robustness when compared to models trained on ImageNet-1k alone (See main table in supplemental attachments).

There are a few prominent strategies for transfer learning in VL-loss models as well; we catalog them below and discuss their strengths and weaknesses.

**Fine-tuning VL models.** Unfortunately, the unique robustness properties of VL-loss models are not conserved when the image tower alone is fine-tuned. As reported in Radford et al. (2021), fine-tuning the VL-loss vision tower using a CE-loss objective improves base accuracy but degrades robustness. This effect grows stronger the longer the model is fine tuned, making fine-tuning the image-tower an inefficient solution for problems where robustness is a consideration.

A similar effect takes place if both vision and language towers are fine-tuned on ground-truth caption data; after 4 epochs of fine tuning on IN1000, a ViT-L-14 CLIP base accuracy improves from .76 to .83; however, average robustness declines from .72 to .69. (See main table in supplemental attachments).

Wise-FT, introduced by Wortsman et al. (2022a) is a fine-tuning method which interpolates the weights of zero-shot CLIP with its fine-tuned counterparts. For certain distribution shifts, it is possible to find a 'sweet spot' where both i.d. and o.o.d. accuracy increase. However, Wise-FT models lose zero-shot capability, and are still not as robust as VL-loss models with the same base accuracy.4

**LiT-tuning.** LiT-tuning, or locked-image text-tuning, is an alternate approach to vision-language training in which a pretrained image tower is aligned with an untrained language model. LiT-tuned models are somewhat more data-efficient than VL models trained from scratch, but they, too, are not as robust as VL-loss models with the same base accuracy. (See 4).

Additionally, we observe the following;

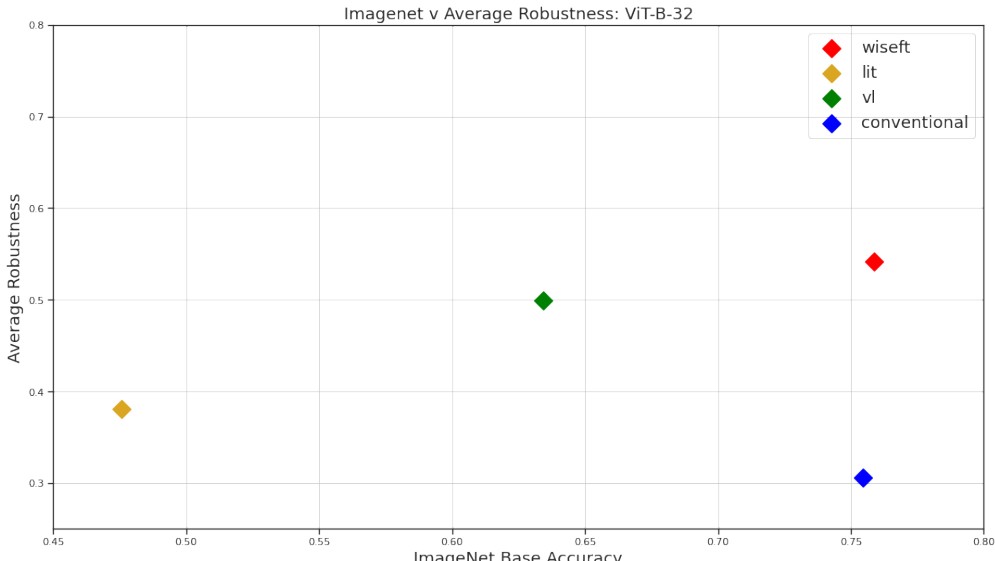

Figure 5: **LiT-tuning on a VL-trained image tower reduces accuracy without altering effective robustness, suggesting that VL pretraining is at least as robust as LiT-tuning.** *Wise-FT tuning greatly increases base accuracy and slightly improves effective robustness, at the cost of zero-shot capability. CE from-scratch training matches Wise-FT accuracy, but sacrifices effective robustness and zero-shot.*

1. Like Wise-FT, LiT-tuning produces models whose i.d. / o.o.d accuracy trade-off fits a line between that of traditional models and VL models – more robust than the former, less robust than the latter. The only exception we found was when we LiT-tuned the vision tower of a ViT trained on the CLIP objective – in this case, LiT-tuning decreased base accuracy while holding effective robustness constant (the near-opposite effect of Wise-FT)

2. LiT-tuning offers negative benefit for fully trained VL models, suggesting that it can only hope to approach, rather than exceed, the accuracy of its baselines (See 5)

3. LiT-tuning performance tends to closely correlate to the base accuracy of the underlying vision model

4. Intriguingly, we find that this is true regardless of the specific dataset used for LiT-tuning – LiT-tuned models trained on small amounts of data are able to recover accuracy on out-of-distribution tasks even when very little data from that distribution shift appears in the pretraining data

5. These experiments suggest that some degree of effective robustness is "locked away" in many vision models, but is lost during the training process, but that certain techniques are able to increase effective robustness disproportionate to the loss in base accuracy, pushing the model 'above the line' we would normally expect. Furthermore, if the distribution shift of interest is known and well-defined, it is possible to select a tuning to optimize for that shift

## B  Additional Findings Regarding Transfer Learning

However, as the commonly used robustness evaluations are themselves derived from ImageNet, it is difficult to know to what extent model performance is dependent on *class-specific ImageNet* pretraining rather than *general* pretraining.

To better understand this distinction, we conduct an ablation study on the pretraining dataset; the results can be found in Tab. 6.

| Pretraining Dataset | Num Pt Classes | Data type | Includes IN100 Classes | IN100 Val. | IN100 V2 | IN100 S | IN100 R | IN100 A | Avg. Rob. | Eff. Rob. Rat. |
|---|---|---|---|---|---|---|---|---|---|---|
| From Scratch | N/A | N/A | N/A | 0.693 | 0.584 | 0.184 | 0.224 | 0.122 | 0.279 | 0.402 |
| JANuS | 100 | Natural | T | 0.698 | 0.588 | 0.208 | 0.237 | 0.128 | 0.29 | 0.416 |
| Fractals | 1000 | Synthetic | F | 0.729 | 0.623 | 0.239 | 0.268 | 0.126 | 0.314 | 0.43 |
| IN1k-minus-IN100 | 1000 | Natural | F | 0.741 | 0.649 | 0.255 | 0.277 | 0.144 | 0.331 | 0.447 |
| IN1k | 1000 | Natural | T | 0.771 | 0.683 | 0.293 | 0.314 | 0.144 | 0.358 | 0.465 |

Table 6: ***Gradations in performance as a result of pretraining dataset choice.*** *Even when holding the size of the pretraining dataset constant, as we do in this experiment, we find that many factors affect the downstream performance of models. The presence of training classes in pretraining data is impactful; in1k-minus-in100 pretraining, where we eliminate the in100 classes, causes a 3% drop in accuracy, compared to the in1k model. Pretraining on a purely synthetic dataset of 1 Mn. fractal images is surprisingly effective, resulting in only a 4% drop in accuracy. JANuS pretraining is the least effective, comparable to training from scratch. This may be attributable to the fact that JANuS has fewer classes, or to the fact that JANuS has more label noise.*

## C    Distribution Shifts

ImageNet is a large-scale visual ontology of images built upon the backbone of the WordNet structure. ImageNet aims to populate the majority of the 80,000 synsets of WordNet with an average of 500–1000 clean and full resolution images, making it a roughly class-balanced, fully supervised dataset. (Deng et al., 2009)

ImageNet-21k, the largest version of ImageNet, contains 14,197,087 images in 21,841 classes.

There now exist a wide range of distribution shifts on ImageNet. These are novel test datasets designed to overcome some of the limitations of the original benchmark. While they cannot remedy issues with the labeling scheme, these datasets do provide challenging new contexts in which to analyze classifier performance.

ImageNet-V2 was designed to duplicate, as closely as possible, the original ImageNet test set. It was intended to answer the question of whether ImageNet-trained classifiers could successfully generalize even to the most mild of distribution shifts. (Recht et al., 2019)

Imagenet-Sketch is a distribution shift covering sketches, paintings, drawings and illustrations of ImageNet classes. This test set is very large and comprehensive. (Wang et al., 2019a)

Imagenet-R is a 200-class subset of ImageNet-2012 focused on renditions of everyday objects, defined broadly as drawings, paintings, photographs of food art, etc. (Hendrycks et al., 2021a)

Imagenet-A is a 200-class subset of ImageNet-2012 which was algorithmically selected – the natural distribution shift captured here is the set of ImageNet-class images which most often fool a RN50. This test is challenging, and tends to include a lot of images with challenges such as occlusion, changes in angle or position, and changes in brightness. (Hendrycks et al., 2021b)

### C.1    Different shifts respond to different interventions

Recent works such as Fang et al. (2022) demonstrate the power of effective robustness as an explanatory tool for performance differences in VL models; Miller et al. (2021) showed that there exists a strong correlation between most models trained on random subsets of a data distribution, and the fully trained model. However, these authors also caution that it has significant limitations – Taori et al. (2020) and Nguyen et al. (2022) show that models trained on more (or different data) can significantly change the effective robustness line of a particular model, and also that these changes were shift-specific, with stronger fits on shifts like ImageNet-V2 and weaker fits on shifts like ImageNet-A.

We find that ImageNet-V2 responds more to model architecture than other shifts, with the handful of non-ResNet models we evaluated outperforming nearly all other models, regardless of training objective.

ImageNet-R and ImageNet-Sketch both showed high sensitivity to the training data, with the CC12M and LAION-15m distributions considerably outperforming even the best YFCC-trained models. These types of shifts are particularly amenable to subset matching strategies.See Fig. 13, Fig. 10 for examples.

On ImageNet-A, CE models significantly underperformed compared to VL models regardless of the data, and all models significantly underperformed compared to the ViT-L CLIP. See Fig. 11 for examples.

| Model Name | Average Validation Accuracy | Average Robust Accuracy |
|---|---|---|
| CLIP-RN50 (1000-class) | 0.5985 | 0.4306 |
| CLIP-RN50 (Avg. 100-class) | 0.8517 | 0.7182 |
| SWSL-RN50 (1000-class) | 0.8362 | 0.6857 |
| SWSL-RN50 (Avg. 100-class) | 0.9524 | 0.7612 |

Table 7: ***Zero-shot model robustness is affected by the difficulty of the task.*** *Both quantity and quality of labels alters model accuracy and robustness under shift; it also changes the comparative performance of VL-loss and CE-loss models. In this table, we transform the 1000-class IN1000 label set into ten 100-class label sets, and find that the resulting predictions are far more accurate and robust, particularly those of the VL-loss model. This finding motivates our choice to study model robustness on multiple label set sizes.*

We also note that there is no readily apparent logit-scaled linear trend in these distribution shifts when one considers models trained on a wide range of different datasets, underscoring the importance of a well-chosen baseline for comparison.

We find that different shifts tend to disadvantage different kinds of models, which makes improving on all of them simultaneously very challenging. The fact that ViT-L CLIP was able to do is both impressive and, given the vital importance of the underlying data distribution in such measures, a mystery which is unlikely to ever be solved. Even the massive public datasets such as LAION are unable to match the performance of the dataset CLIP was trained on, although other factors might possibly have played a role.

A standardized benchmark of distribution shifts on ImageNet would be a welcome contribution to this area of research.

# D   Pretraining Datasets

Today, many SOTA models are pretrained on web-scale unsupervised data. We utilized three such datasets in our experiments. We observe that one major challenge of conducting research on unsupervised datasets is that the links provided as part of the dataset fail more and more over time, leading to each group getting a different version of the dataset. Therefore, to the extent possible, we report the details of each dataset in the appendix, and encourage other researchers working with these datasets to do the same.

CC-12M is a lightly supervised web-scale dataset created by Google. The image-caption pairs in CC-12M were filtered and selected for the purposes of training models to caption images. (Changpinyo et al., 2021) Our version of CC12M contained 9703885 image-caption pairs.

YFCC-15M is a subset of YFCC-100M, which is 100M image-metadata pairs taken from Yahoo-Flickr in 2016. The subset was selected by OpenAI. This dataset contains images and metadata, which includes a "title" and a "description" field. These fields are combined and processed in various ways by researchers in order to generate captions for models to train on. (Thomee et al., 2016) Our version of YFCC contained 14825134 image-caption pairs.

LAION is a 5B image-caption dataset recently created by LAION.ai. It is the first publicly available dataset which matches the scale of the datasets used by the large companies to train their best models. (Schuhmann et al., 2021a) The subset of LAION we refer to as LAION-15m contained 13775512 image-caption pairs.

# E   Data Quality: Other Considerations

There are various other important considerations, aside from the raw count of per-class samples, in determining the utility of a dataset. We follow Santurkar et al. (2022) in referring to these as **data quality** considerations.

We proceed to discuss some of these considerations.

### E.1 Image Size

In Fig. 12 (R), we plot average robustness against image resolution, expressed as the ratio of actual model resolution to maximum model resolution in the study (800px). We find that increasing input image resolution leads to gains in robustness on IN1000 and IN100-Dogs, but that these effects are smaller than choice of architecture and number of model parameters.

### E.2 Class Imbalance

Aside from simply considering the overall *size* of a dataset, it is important to also consider the *per-class size* of a dataset.

The significance of this distinction can easily be seen on JANuS, when we compare the performance of OI100-trained and IN100-trained models in Tab. 9.

Despite the fact that OI100 is a slightly larger than IN100, models trained on OI100 perform worse on IN100-Val than models trained on IN100. We find that the extreme class imbalance shown in Fig. E.3 is the cause of most, but not all, of the decrease in accuracy.

VL-loss class imbalances (detected by searching for exact-match classnames in caption strings) are also present in the other web-scraped datasets in JANuS, LAION and YFCC; this may contribute to the lower performance of VL-loss models on long-tailed classification.

### E.3 Label Set Size

One important, but rarely considered, factor in distributional robustness is the size of the label set.

Specifically, we find that models trained on many-class problems become more accurate and robust when the label set size is reduced to a subset of those classes at inference time, and the improvements are not necessarily proportionate

Very large label sets have previously been shown cause declines in base accuracy. (Mohammed & Umaashankar, 2018) In Tab. 7, we show that this effect is present even when only 1000 classes are used (the size of ImageNet), and that it affects distributional robustness as well as base accuracy.

Based on this observation, we recommend taking into account label set size when evaluating model robustness, and propose reducing the size of the label set as an effective intervention for improving robustness.

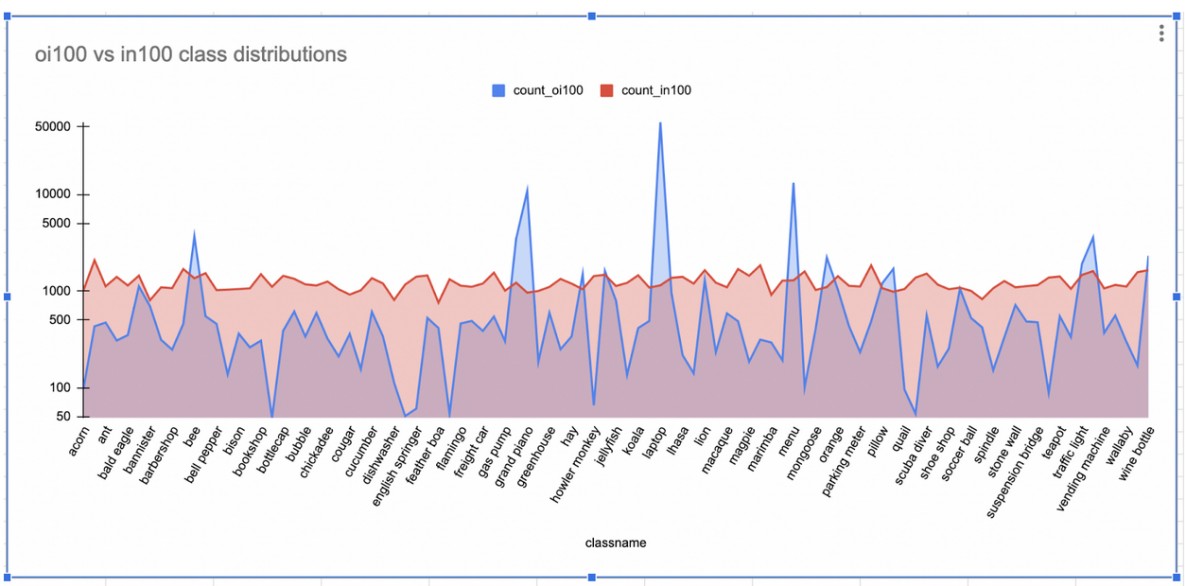

Figure 6: This log-scale figure shows the extreme class imbalance of the *unfiltered* OI100 dataset, compared to the *prefiltered* IN100 dataset; certain classes which are very common in web-scraped images, such as laptops, are overrrepresented, while others are not represented at all. The OI100 class imbalance is produced by a difference in dataset labeling strategies. VL-loss class imbalances (detected by searching for exact-match classnames in caption strings), which are present in the other web-scraped datasets in JANuS, LAION and YFCC, co-occur with comparatively low accuracy scores on fine-grained classification tasks.

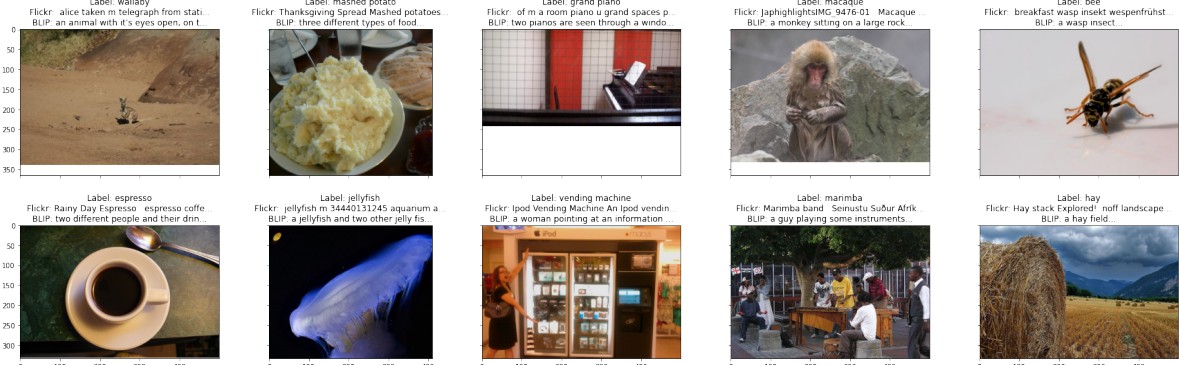

Figure 7: **ImageNet-100 samples from JANuS.**

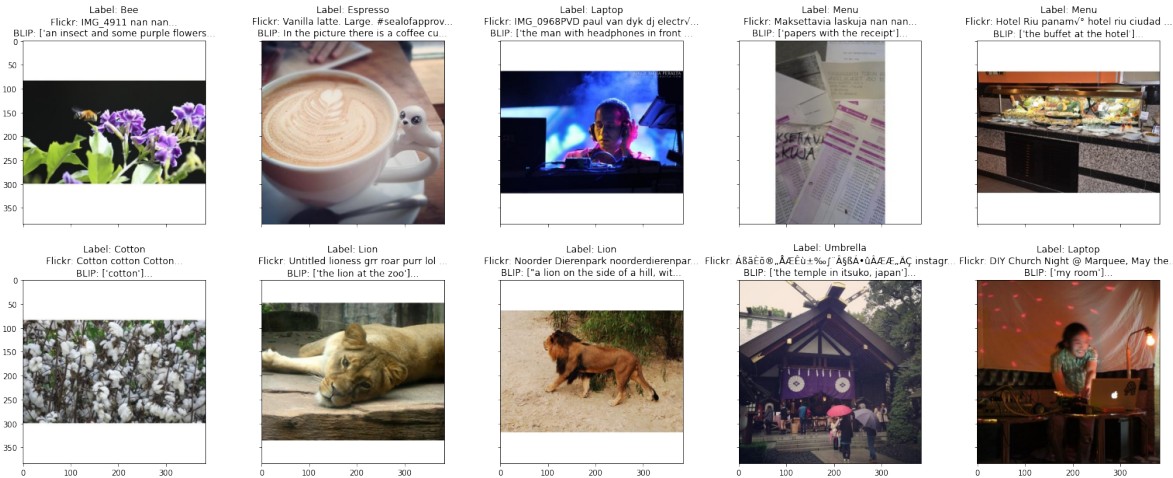

Figure 8: **OpenImages-100 samples from JANuS.**

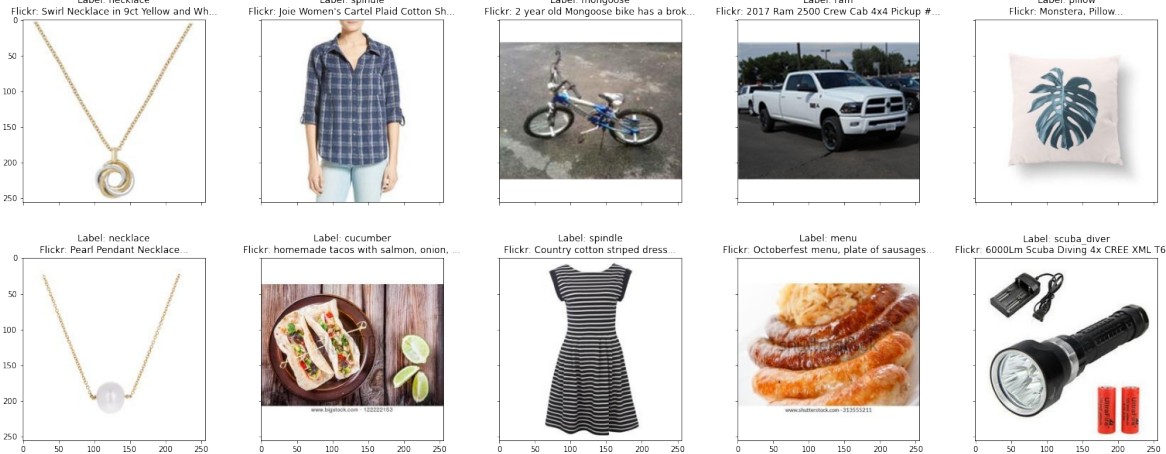

Figure 9: **LAION-100 samples from JANuS.**

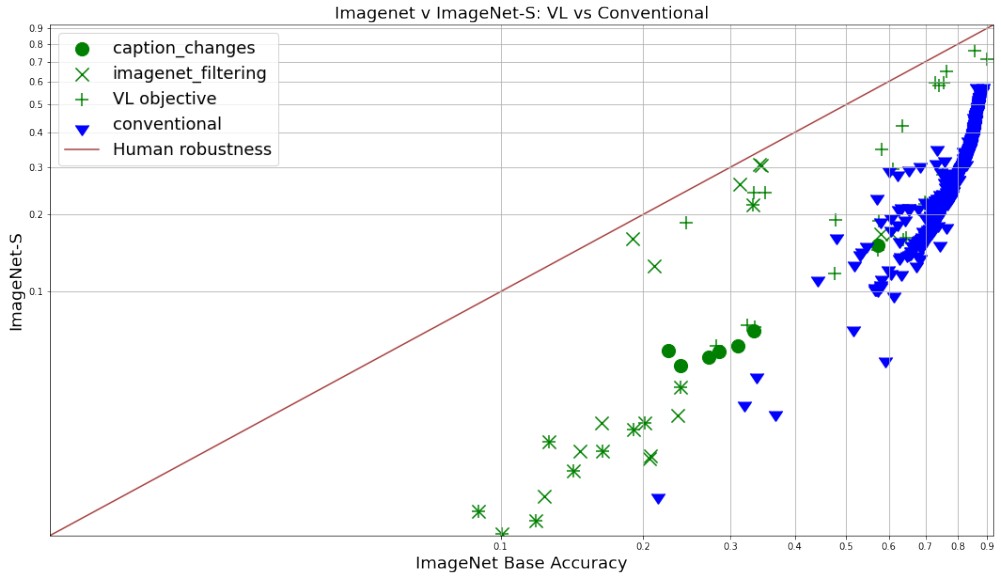

Figure 10: **Non-linearities in ImageNet-Sketch.** *ImageNet-sketch performance is not linear, with only the very largest VL models showing a reliable improvement over CEly trained models, when controlling for dataset size.*

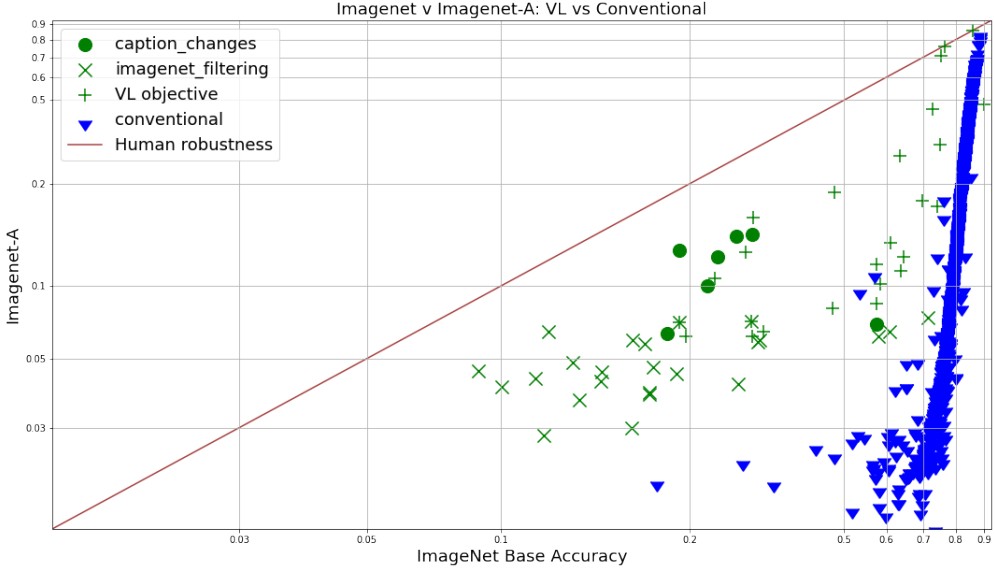

Figure 11: **ImageNet-A is learnable by all models at extremely high base accuracy.** *Although VL models seem to learn ImageNet-A faster than CE models, CE models reach near-parity with VL models when base accuracy gets very high.*

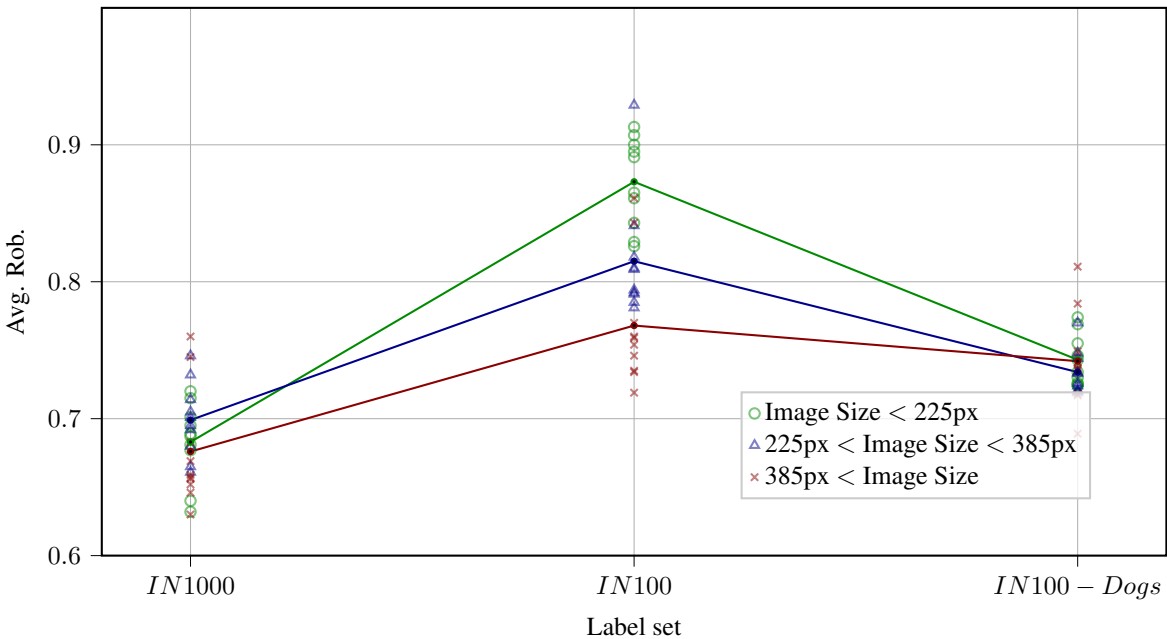

Figure 12: **Robustness effects of scaling input image resolution.** *Previous results from Tan & Le (2019) have shown that image resolution is an important factor in base model accuracy. Our meta-analysis indicates that input image resolution can have a strong positive effect on robustness as well.*

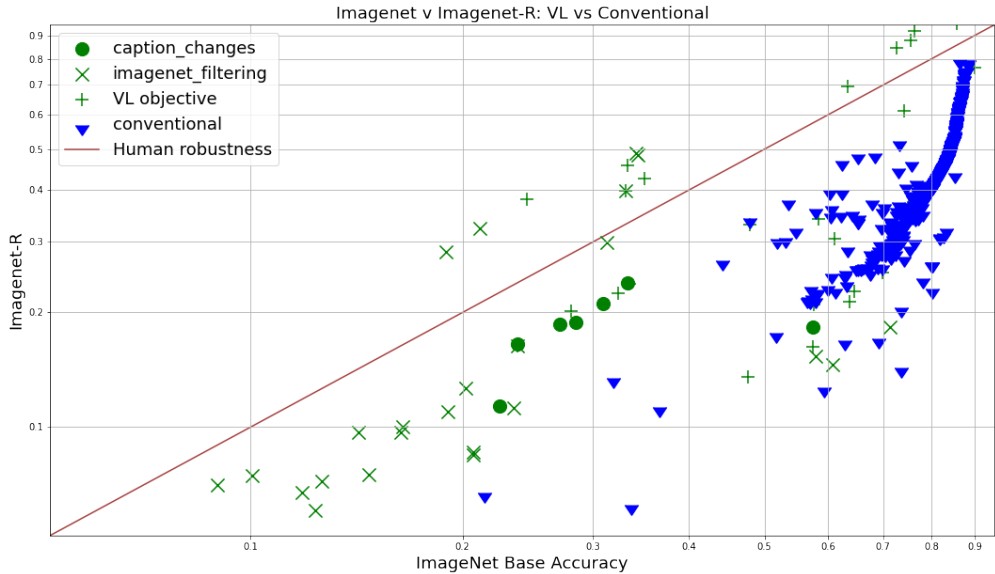

Figure 13: **VL performance on ImageNet-R outstrips base accuracy.** *On ImageNet-R, which is a 200-class subset of ImageNet, VL models are able to achieve higher accuracy than on ImageNet itself. VL continues to outperform CE models on this dataset, even at very high accuracies.*

| in1k classname | search term | good samples | total samples | avail. samples | pct. good |
|---|---|---|---|---|---|
| lion | lion | 962 | 1000 | 450000 | 0.96 |
| wine bottle | wine bottle | 925 | 1000 | 29500 | 0.93 |
| book shop | bookstore | 816 | 984 | 83000 | 0.83 |
| parking meter | parking meter | 377 | 1000 | 9500 | 0.38 |
| african elephant | african elephant | 885 | 1000 | 44000 | 0.89 |
| bagel | bagel | 699 | 988 | 20500 | 0.71 |
| tarantula | tarantula | 667 | 981 | 9000 | 0.68 |
| ice cream | ice cream | 741 | 984 | 154500 | 0.75 |
| fig | fig | 517 | 1000 | 46000 | 0.52 |
| shoe shop | shopping shoes | 425 | 1000 | 13000 | 0.43 |
| french bulldog | french bulldog | 887 | 996 | 7500 | 0.89 |
| hen | hen | 412 | 1000 | 73000 | 0.41 |
| guacamole | guacamole | 683 | 998 | 6500 | 0.68 |
| broccoli | broccoli | 679 | 997 | 19000 | 0.68 |
| howler monkey | howler monkey | 817 | 847 | 9000 | 0.96 |
| scuba diver | scuba diver | 827 | 1000 | 15000 | 0.83 |
| spindle | "spindle wool, spindle -wool thread" | 311 | 867 | 867 | 0.36 |
| lhasa | lhasa dog | 719 | 1000 | 2500 | 0.72 |
| traffic light | stoplight | 622 | 991 | 5500 | 0.63 |
| lionfish | lionfish | 552 | 897 | 6500 | 0.62 |
| popsicle | popsicle -animal -sticks -animals -insect -insects -icicle -garden -sticks -icicles -gardens -toes -label -labels | 638 | 943 | 7500 | 0.68 |
| lampshade | lampshade | 446 | 807 | 6500 | 0.55 |
| spiderweb | spiderweb -spiderman -halloween -pumpkin -butterfly -pleiades -nebula -stars | 832 | 996 | 17500 | 0.84 |
| lifeboat | lifeboat | 572 | 1000 | 13000 | 0.57 |
| cucumber | cucumber -sea -spider -beetle -flower -spiral | 730 | 999 | 26500 | 0.73 |
| english springer | english springer spaniel | 772 | 993 | 3500 | 0.78 |
| macaw | macaw | 972 | 1000 | 13500 | 0.97 |
| mailbox | mailbox | 900 | 1000 | 36500 | 0.9 |
| peacock | peacock -butterfly | 966 | 999 | 72000 | 0.97 |
| bee | bumblebee OR wasp OR hornet -jet -airplane -helicopter -navy -aircraft -comic -RIAT -military -Helicopter -Helicopters -helicopters -aviation -Hudson -car -basketball -sports -Transformers -cosplay -disfrazado -costume -transformer AND flower | 686 | 761 | 110000 | 0.9 |
| dungeness crab | dungeness AND crab -restaurant -breakfast -lunch -dinner -shack -creels -traps -cannery | 474 | 1000 | 1500 | 0.47 |

| | | | | | |
|---|---|---|---|---|---|
| banana | banana -plant -blossom -flower -seed -seedlings -tree -spider -leaf -abstract -bay -band -festival -doll -sexy -sexiest -bread -soup -puree -smoothie -car -plantation -cake -cream -monkey -pudding -zoo -republic -boxes -buying -selling -vendor -bridge -scone -moon | 793 | 995 | 65000 | 0.8 |
| corn | corncob | 354 | 1000 | 1000 | 0.35 |
| lemon | lemon -plant -blossom -flower -seed -seedlings -tree -spider -leaf -abstract -bay -band -festival -doll -sexy -sexiest -bread -soup -puree -smoothie -car - plantation -scent -fresh -cleaner -butterfly -grove -shots -car -sunrise -paint - graffiti -origami -cake -cream -pudding -boxes -buying -selling -vendor -bridge -scone -don -lime | 693 | 1000 | 65000 | 0.69 |
| marimba | marimba instrument | 127 | 283 | 283 | 0.45 |
| orange | orange food fruit -plant -blossom -flower -seed - seedlings -tree -spider -leaf -abstract -bay -band -festival -doll -sexy -sexiest -bread -soup -puree -smoothie -car -plantation -cake -cream -monkey -pudding -zoo -republic -boxes -buying -selling -vendor -bridge -scone -moon -cupcake -cake -sales -seller -pancakes -crepes -crep -crepe -pancake -cookie -flavored -juice -soda -pop -beach -island -cove -grove -street -drive -tea -curd -marmalade -bars -cabs -chicken -cheesecake -pie -milk | 744 | 1000 | 4000 | 0.74 |

| | | | | | |
|---|---|---|---|---|---|
| bell pepper | bell pepper vegetable -plant -blossom -flower -seed -seedlings -tree -spider -leaf -abstract -bay -band -festival -doll -sexy -sexiest -bread -soup -puree -smoothie -car -plantation -cake -cream -monkey -pudding -zoo -republic -boxes -buying -selling -vendor -bridge -scone -moon -cupcake -cake -sales -seller -pancakes -crepes -crep -crepe -pancake -cookie -flavored -juice -soda -pop -beach -island -cove -grove -street -drive -tea -curd -marmalade -bars -cabs -chicken -cheesecake -pie -milk -market -spice | 392 | 505 | 505 | 0.78 |
| espresso | espresso coffee -maker -machine -beans -building -exterior -window | 828 | 1000 | 22000 | 0.83 |
| mashed potato | mashed potato | 635 | 996 | 10000 | 0.64 |
| stingray | stingray water -dolphin -shark -cruise -boat -scuba -fish | 600 | 983 | 2000 | 0.61 |
| flagpole | flagpole -lighthouse -church -bank -station | 614 | 991 | 7000 | 0.62 |
| teapot | teapot -tea -flower -tower -building -dome -art -fashion -vase -store -stores -shop -shops -Sagittarius -project365 -fountain -candle -mug -teacup -keg -vessel -amphora -urn -coffeepot | 660 | 997 | 10500 | 0.66 |
| umbrella | umbrella | 911 | 1000 | 126000 | 0.91 |
| beer bottle | beer bottle -house -door -brewery -glass -cap | 909 | 1003 | 19000 | 0.91 |
| barn | barn -swallow -owl -bird | 980 | 1000 | 115000 | 0.98 |
| christmas stocking | christmas stocking fireplace | 317 | 779 | 779 | 0.41 |
| magpie | magpie -screenshots -moth -butterfly -coprinopsis -thieving -mushroom | 736 | 983 | 25500 | 0.75 |
| mitten | mitten glove | 800 | 995 | 1500 | 0.8 |
| ram | ram sheep -Church -window -Window -church -school -dance -parade -festival -celebration -festivities -community -fair -ewe -fox -lamb -bird -cat -dog -Dodge | 742 | 1000 | 3000 | 0.74 |

| | | | | | |
|---|---|---|---|---|---|
| warthog | warthog animal -zebra -cheetah -leopard -giraffe -gazelle -hippo -rhino -donkey -armadillo -elephant -crocodile -lion -leopard -impala -cat -monkey -bird | 946 | 997 | 2500 | 0.95 |
| goose | geese | 474 | 500 | 69000 | 0.95 |
| bubble | soap bubble -dancer -dance -fairy -tree -leaf -leaves -flowers -water -toy -art -abstract -museum -dog -cat -butterfly -food -wine -beer -chocolate -Chocolate | 414 | 500 | 5000 | 0.83 |
| cougar | cougar animal -warthog -mascot -zebra -cheetah -leopard -giraffe -gazelle -hippo -rhino -donkey -armadillo -elephant -crocodile -lion -leopard -impala -cat -monkey -bird -lake -Lake -river -River -blonde -Blonde -woman -girl -milf -bear -cliff -Cliffs -cliffs -military -wallaby -horse -jet -print | 297 | 500 | 1000 | 0.59 |
| daisy | daisy flower | 500 | 500 | 52000 | 1 |
| menu | menu | 431 | 500 | 92000 | 0.86 |
| bald eagle | bald eagle | 475 | 500 | 33500 | 0.95 |
| necklace | necklace jewelry -brooch -pendant -creation -earring -earrings -bracele -ring -Engraver -bauble -anklet | 478 | 500 | 12500 | 0.96 |
| chickadee | chickadee bird -Goldfinch -goldfinch -robin -thrush -jay -cardinal -woodpecker -wren -hawk -raven -titmouse -nuthatch | 494 | 500 | 9000 | 0.99 |
| stone wall | """stone wall""" | 424 | 500 | 32000 | 0.85 |
| flamingo | flamingo bird | 476 | 500 | 38500 | 0.95 |
| gas pump | gas station | 348 | 500 | 41000 | 0.7 |
| vulture | vulture bird -hawk -crow -eagle | 489 | 500 | 15500 | 0.98 |
| pizza | """pizza pie"" -Fest -festival -summit -experience -party -band -moon -parade -Parade -harvard -mosaic -montage" | 305 | 500 | 1000 | 0.61 |

| | | | | | |
|---|---|---|---|---|---|
| wallaby | wallaby -warthog -mascot -zebra -cheetah -leopard -giraffe -gazelle -hippo -rhino -donkey -armadillo -elephant -crocodile -lion -leopard -impala -cat -monkey -bird -koala -sports -kangaroo -soccer -football -food -church -hills -stadium -tribute -grass -rugby -apartment -car | 369 | 500 | 10000 | 0.74 |
| hay | haystack field -hole -trail -poster -sign | 360 | 500 | 1000 | 0.72 |
| grand piano | "kawai grand piano, steinway grand piano" | 312 | 455 | 455 | 0.69 |
| laptop | laptop | 443 | 500 | 98000 | 0.89 |
| dishwasher | dishwasher appliance | 191 | 268 | 268 | 0.71 |
| cricket | cricket -batting -sports -team -match | 337 | 500 | 44000 | 0.67 |
| sea slug | nudibranch | 468 | 500 | 12500 | 0.94 |
| mongoose | mongoose -bike -bicycle -park -tree -joe -rocket -military -airplane -toy -car | 379 | 500 | 5000 | 0.76 |
| siamese cat | siamese cat -bangkok -flower -snake -campaign -wat -costume -cosplay -festival | 416 | 500 | 13000 | 0.83 |
| freight car | freight car | 491 | 500 | 70500 | 0.98 |
| vending machine | """vending machine""" | 411 | 500 | 13000 | 0.82 |
| bottlecap | bottlecap -tab | 448 | 500 | 3500 | 0.9 |
| acorn | acorn -woodpecker -fairy -squirrel -weevil -travel -squash -street | 352 | 500 | 25000 | 0.7 |
| feather boa | feather boa | 135 | 500 | 2000 | 0.27 |
| macaque | macaque | 485 | 500 | 14500 | 0.97 |
| bolete | boletus | 444 | 500 | 3500 | 0.89 |
| border terrier | """border terrier""" | 422 | 500 | 1500 | 0.84 |
| barbell | barbells | 352 | 500 | 1000 | 0.7 |
| fly | housefly | 398 | 500 | 1500 | 0.8 |
| suspension bridge | suspension bridge | 432 | 500 | 33500 | 0.86 |
| jellyfish | jellyfish | 477 | 500 | 46500 | 0.95 |
| barbershop | barbershop -quartet -singers | 430 | 500 | 9000 | 0.86 |
| koala | koala | 458 | 500 | 32500 | 0.92 |
| bannister | bannister staircase | 174 | 183 | 183 | 0.95 |
| pillow | pillow -talk -fight -cat -dog -moss -sky -cloud -sky | 420 | 500 | 34500 | 0.84 |
| bib | baby bib -shower -food | 406 | 500 | 1500 | 0.81 |
| junco | junco bird -finch -sparrow -thrush -cardinal -woodpecker -jay | 475 | 500 | 7000 | 0.95 |
| chainlink fence | chainlink fence | 375 | 500 | 3500 | 0.75 |
| soccer ball | """soccer ball"" -match -game -milky -beach -Lewes" | 349 | 500 | 2500 | 0.7 |
| stupa | stupa | 418 | 500 | 23500 | 0.84 |

| quail | quail bird -finch -sparrow -thrush -cardinal -woodpecker -jay -partridge -rabbit -hawk -avocet -deer -dog -wolf -coyote -gopher -eagle -vole -molerat -butterfly | 396 | 500 | 11000 | 0.79 |
|---|---|---|---|---|---|
| padlock | padlock | 378 | 500 | 9500 | 0.76 |
| great white shark | """great white shark""" | 309 | 500 | 2000 | 0.62 |
| totem pole | """totem pole"" wood" | 383 | 500 | 1000 | 0.77 |
| ant | ant insect | 447 | 500 | 18000 | 0.89 |
| bison | bison | 429 | 500 | 41500 | 0.86 |
| greenhouse | greenhouse | 407 | 500 | 82000 | 0.81 |

Table 8: **JANuS Supervision: Search Terms and Sample Quality** *Since many of the findings in our paper highlight the importance of both the amount and type of label noise, this table records statistics pertaining to our filtration process for the new samples in IN100. In the search term field, a - symbol indicates that all samples which included that word in the title, tags or description were NOT matched. Boolean OR, AND, and "" symbols behave as they typically do.*

**Adding BLIP Captions to JANuS**

Since we could not find human-authored captions for ImageNet, we used BLIP Li et al. (2022a) to generate descriptive captions on ImageNet-100. BLIP often uses word fragments to describe objects, so we used a spell checker as a simple intervention to improve the quality of BLIP captions. Finally, because BLIP's vocabulary does not include many of the specialized classes available in ImageNet, we augmented the BLIP captions with Flickr image titles, the form of text which is most commonly available for an image. We used top p=0.9, max length=40, min length=5, repetition penalty=1.1.

We repeated the process for OpenImages-100. However, we used human-authored captions sourced from Pont-Tuset et al. (2020) instead of BLIP whenever available; around 16,000 out of the 135,000 OpenImages-100 samples had human-authored captions.

# F Classwise Shifts

## F.1 Per class accuracies for CLIP RN50 and SWSL RN50

In the supplementary attachments (2_clip_rn50_per_class_acc, 3_swsl_rn50_in1000_conf_mat), we provide per-class confusion matrices on IN1000 for CLIP ResNet-50, trained on 400 Mn samples, as well as a semi weakly supervised ResNet-50 trained by Facebook on 1 Bn samples.Yalniz et al. (2019)

In addition to classnames which are literally identical (there are two instances of the class "missile" and two instances of the class "sunglasses" in the OpenAI classnames for IN1000), we find that the model struggles to disambiguate short words with similar starting token strings, such as "quail", "quilt" and "quill", and classes that start with common (and contextually misleading) words, such as "night snake".

# G Approach 2: Model details and results

**Results**

The complete results for approach 2 are available as part of our supplementary attachment (1_caption-net_in1k_model_results_and_metadata).

**Model details: timm**

Our meta-analysis made extensive use of the popular timm Wightman (2019) computer vision library, including models from Zhang et al. (2021); Bao et al. (2022); Kolesnikov et al. (2019); Srinivas et al. (2021); Touvron et al. (2021); Xu et al. (2021); Dai et al. (2021); d'Ascoli et al. (2021); Touvron et al. (2022); Huang et al. (2016); Yu et al. (2017); Chen et al. (2017); Maaz et al. (2022); Li et al. (2022b); Xie et al. (2020); Tan & Le (2019); Wu et al. (2018); Han et al. (2020); Wang et al. (2019b); Vaswani et al. (2021); Graham et al. (2021); for details about any of the timm models, please refer directly to the timm repository. In our supplementary results spreadsheet, the name field for each model is the same as that model's name in the timm repository – where model names have been modified between the time our evaluations took place and the time the paper was completed, we note the new names in the updated name column.

**Model details: other**

For non-timm models, we include a key for the terms used to describe them in the meta-analysis:

CLIP: denotes vision language pretraining using the CLIP objective

RN101: A ResNet-101 model from He et al. (2015).

RN50: A ResNet-50 model from He et al. (2015).

yfcc: The yfcc dataset from Thomee et al. (2016).

cc12m: The CC-12m dataset. (Changpinyo et al., 2021)

laion: the LAION dataset from Schuhmann et al. (2021b).

WIT-400m: The original dataset used to train CLIP models by Radford et al. (2021).

no-imagenet-classnames: All samples whose caption contained an ImageNet classname were filtered out of the dataset using subset matching, as described in Sec. H.

RN50x64: A 64x scaling of the ResNet architecture according to the EfficientNet scaling rule, introduced by Radford et al. (2021).

LiT: LiT tuning as described by Zhai et al. (2021).

# H    Subset matching

For unsupervised web-scraped captioned datasets (such as LAION and YFCC), ground-truth class labels do not exist. Therefore, we must choose a strategy to assign class labels to samples in such datasets. VL-loss models use captions as labels. There is no easy way for CE-loss models to directly use captions as labels. To facilitate this, we propose a strategy we call *subset matching*, a modification of the "substring matching" technique proposed by Fang et al. (2022).

This strategy, illustrated in detail in 14, labels samples as follows. First, construct a dict of integers and "matching terms". A matching term is a string judged to be a good text representation of an image class, such as the string 'elephant' for an image of an elephant. Our standard choice of matching terms is based on Radford et al. (2021).

If a sample caption contains a matching term, then the corresponding integer class label is applied. If the sample caption contains multiple matching terms, then we apply one of three strategies, which we label strict, multi-class (mc) and single-class (sc) matching, explained in detail in Sec. H; we use single-class matching whenever possible, since it usually performs best. If the sample caption contains no matching terms for any class, then no label is applied and the image is dropped from the training set. Otherwise, the caption is replaced with the corresponding integer-valued label.

A **subset matching strategy** is an algorithmic method for applying machine labels to images, based on caption labels. All of these methods share in common the same underlying approach, as seen in Fig. 14.

In this section, we fully define and describe some important variations on the basic subset matching strategy as described in the main paper. All of our subset matching experiments utilized one of these three strategies.

**Strict:** Strict subset matching means that the model only applies the label to the image if the caption contains term(s) which map to exactly one class.

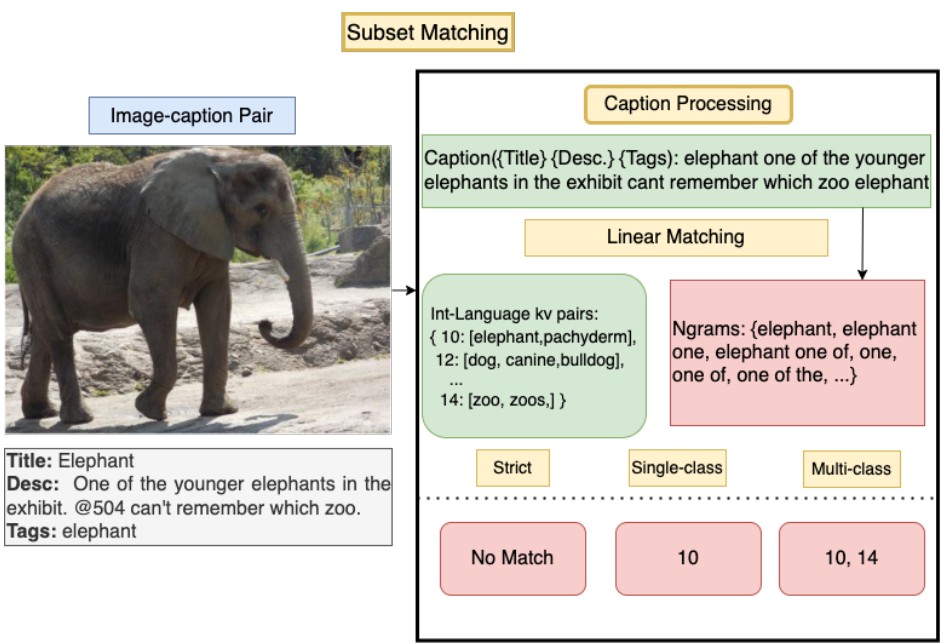

Figure 14: **Subset matching; an overview.** *Subset matching is a simple labeling strategy for unsupervised image-caption pairs. The caption is processed and converted to n-grams which are then matched against a database of terms which point to integer-label classes.*

Strict subset matching was generally the most accurate method on ImageNet – we believe this is because of the ImageNet dataset filtration strategy, in which label selection is strongly dependent on caption contents.

It was generally less accurate method than single-class on OpenImages, where labels and caption contents are independently derived.

We find that strict matching performance tends to degrade when the pool of matching terms grow; it also tends to punish synthetic captions, which use a smaller vocabulary than web-scraped or annotated captions.

**Single class:** In single-class subset matching, the model greedily takes the first matching term to be the true class and ignores all subsequent matching terms.

As a general matter, we found that single-class matching struck the best balance between dataset utilization and accuracy, and we used this method for most of our experiments.

**Multi class:** In multi-class subset matching, we match up to 25 classes per sample (if we see multiple terms for a single class, we ignore those additional terms, and we do not attempt to rank classes by frequency).

The cross-entropy loss of the model is then given by the sum of the loss on each class; in other words, we reward the model for applying a high probability on each label assigned to the sample and for applying a low probability to each label which was not assigned to the sample.

This approach, while intriguing, was challenging because we only had one ground-truth label for each image; therefore, multi-class matching was always less accurate than single-class matching in direct comparison.

Since our cross-entropy model used a softmax loss, we found that model error tended to be high as the number of matched classes grew. We also found empirically that images which actually required multiple labels were not particularly common in our dataset. Perhaps for these reasons, this approach performed worse than single-class matching in most experiments.

**Additional term definitions.**

*Label accuracy.* On datasets for which supervised ground-truth labels exist, we report label accuracy (Label acc.) as the count of machine-generated labels which match ground truth labels, divided by the total number of samples in the dataset.

*Dataset utilization.* Dataset utilization (Ds. Util) of a model on a dataset is the ratio of correctly labeled samples to total number of samples (including correct, incorrect and unlabeled samples). We use this metric to judge how useful a labeling strategy is; ground truth labels have a utilization of 100%; automated labeling methods gives typically significantly less utilization.

# I  Approach 1 Results

Please see Tab. 9 for the results for models trained using approach 1 (JANuS). Baseline result for each constituent dataset is in **bold**.

**RESULTS KEY**

*imbal*: Class-imbalanced dataset

*384-res*: Trained at 384 image resolution

*RN50x4*: Used RN50x4 backbone

*RN26*: Used RN26 backbone

*int*: Trained using CE-loss

*VL*: Trained using VL-loss

*jpeg10*: JPEG compression of images, strength 10

*cliplabel*: Trained using noisy integer labels generated by a CLIP ViT-L-14 model

*swinlabel*: Trained using noisy integer labels generated by a Swin transformer

*ttd*: VL model trained using noisy captions (title, tag, description from Flickr where available, alt-text where not available)

*tags*: VL model trained using tags only

*title*: VL model trained using title only

*sbm*: Samples selected using subset matching

*size*: Sample size was size-controlled to be identical to supervised dataset size

*gtcaps*: VL model trained on ground-truth captions generated from human labels

*tokscramble*: VL model trained with the order of tokens randomly scrambled during training and inference

*tokstrip*: VL model trained with all tokens not used for inference stripped during training

*blipcap*: Captions generated using a BLIP captioning model

*ofa*: Captions generated using an OFA captioning model

*annotcap*: Human descriptive annotations for images (available only for a subset of OI100)

*classname-only*: Captions are classname and nothing else

*classbal*: Corrected for class imbalance prior to training

| Model Name | IN100-Val | IN100-V2 | IN100-S | IN100-R | IN100-A | Avg. Rob. | Eff. Rob. | Label Acc. | DS Util. |
|---|---|---|---|---|---|---|---|---|---|
| **Data Budget: 1/8x** | | | | | | | | | |
| **in100(1/8th)-int** | 0.618 | 0.485 | 0.139 | 0.189 | 0.104 | 0.229 | 0.371 | 1 | 1 |
| in100(1/8th)-VL-gtcaps | 0.577 | 0.461 | 0.17 | 0.196 | 0.104 | 0.233 | 0.402 | 1 | 1 |
| **Data Budget: 1x (bal)** | | | | | | | | | |
| **in100-int** | 0.87 | 0.791 | 0.373 | 0.378 | 0.153 | 0.424 | 0.487 | 1 | 1 |
| in100-int-384-res | 0.841 | 0.768 | 0.337 | 0.343 | 0.177 | 0.406 | 0.483 | 1 | 1 |
| in100-int-RN26 | 0.81 | 0.736 | 0.377 | 0.363 | 0.135 | 0.403 | 0.498 | 1 | 1 |
| in100-int-RN50x4 | 0.874 | 0.805 | 0.336 | 0.369 | 0.19 | 0.425 | 0.486 | 1 | 1 |
| in100-int-ViT-S-16 | 0.713 | 0.584 | 0.124 | 0.204 | 0.11 | 0.256 | 0.359 | 1 | 1 |
| in100-int-DeIT-S-16 | 0.756 | 0.643 | 0.164 | 0.222 | 0.137 | 0.292 | 0.386 | 1 | 1 |
| in100-int-GCViT-S-16 | 0.803 | 0.702 | 0.378 | 0.356 | 0.143 | 0.395 | 0.492 | 1 | 1 |
| in100-int-jpeg10 | 0.809 | 0.728 | 0.341 | 0.345 | 0.131 | 0.386 | 0.478 | 1 | 1 |
| in100-int-cliplabel | 0.813 | 0.717 | 0.278 | 0.328 | 0.127 | 0.363 | 0.446 | 0.9 | 1 |
| in100-int-size-sbm-ttd | 0.801 | 0.7 | 0.267 | 0.311 | 0.124 | 0.351 | 0.438 | 0.89 | 1 |
| in100-int-sbm-ttd | 0.754 | 0.674 | 0.285 | 0.331 | 0.123 | 0.353 | 0.468 | 0.89 | 0.72 |
| in100-int-sbm-tags | 0.723 | 0.636 | 0.251 | 0.297 | 0.109 | 0.323 | 0.447 | 0.87 | 0.58 |
| in100-int-sbm-title | 0.686 | 0.603 | 0.237 | 0.301 | 0.107 | 0.312 | 0.455 | 0.94 | 0.49 |
| in100-VL-gtcaps | 0.849 | 0.768 | 0.37 | 0.373 | 0.17 | 0.421 | 0.495 | 1 | 1 |
| in100-VL-gtcaps-tokscramble | 0.837 | 0.765 | 0.372 | 0.399 | 0.162 | 0.425 | 0.507 | 1 | 1 |
| in100-VL-jpeg10 | 0.75 | 0.682 | 0.311 | 0.352 | 0.144 | 0.372 | 0.496 | 1 | 1 |
| in100-VL-ttd | 0.587 | 0.487 | 0.162 | 0.173 | 0.085 | 0.227 | 0.386 | 0.89 | 0.72 |
| in100-VL-ttd-tokstrip | 0.585 | 0.475 | 0.145 | 0.19 | 0.081 | 0.223 | 0.381 | 0.89 | 0.72 |
| in100-VL-blipcap | 0.405 | 0.351 | 0.138 | 0.165 | 0.07 | 0.181 | 0.447 | 0.61 | 0.28 |
| in100-VL-classname-only | 0.236 | 0.218 | 0.122 | 0.1 | 0.05 | 0.123 | 0.521 | 1 | 1 |
| **Data Budget: 1x (imbal)** | | | | | | | | | |
| **oi100-int** | 0.667 | 0.595 | 0.316 | 0.399 | 0.156 | 0.367 | 0.549 | 1 | 1 |
| oi100-int-classbal | 0.812 | 0.734 | 0.39 | 0.399 | 0.167 | 0.423 | 0.520 | 1 | 1 |
| oi100-int-cliplabel | 0.631 | 0.553 | 0.273 | 0.343 | 0.134 | 0.326 | 0.516 | 0.9 | 1 |
| oi100-int-sbm-ttd | 0.369 | 0.304 | 0.109 | 0.2 | 0.104 | 0.179 | 0.486 | 0.48 | 0.08 |
| oi100-VL-gtcaps | 0.694 | 0.644 | 0.35 | 0.423 | 0.177 | 0.399 | 0.574 | 1 | 1 |
| oi100-VL-ttd | 0.26 | 0.22 | 0.065 | 0.121 | 0.066 | 0.118 | 0.454 | 0.53 | 0.11 |
| oi100-VL-blipcap+annotcap | 0.343 | 0.291 | 0.09 | 0.174 | 0.055 | 0.152 | 0.443 | 0.46 | 0.14 |
| oi100-VL-blipcap | 0.298 | 0.28 | 0.095 | 0.151 | 0.065 | 0.148 | 0.495 | 0.42 | 0.12 |
| **Data Budget: 2x** | | | | | | | | | |
| **in100+oi100-int** | 0.904 | 0.829 | 0.489 | 0.515 | 0.213 | 0.512 | 0.566 | 1 | 1 |
| in100+oi100-int-384-res | 0.878 | 0.807 | 0.442 | 0.455 | 0.241 | 0.486 | 0.554 | 1 | 1 |
| in100+oi100-int-ViT-S-16 | 0.785 | 0.667 | 0.211 | 0.277 | 0.146 | 0.325 | 0.414 | 1 | 1 |
| in100+oi100-int-RN26 | 0.858 | 0.771 | 0.463 | 0.465 | 0.204 | 0.476 | 0.555 | 1 | 1 |
| in100+oi100-int-RN50x4 | 0.903 | 0.838 | 0.458 | 0.478 | 0.229 | 0.501 | 0.555 | 1 | 1 |
| **Data Budget: 4x** | | | | | | | | | |
| **in100+laion100-int** | 0.901 | 0.827 | 0.614 | 0.636 | 0.201 | 0.57 | 0.633 | N/A | N/A |
| in100+laion100-int-ViT-S-16 | 0.787 | 0.675 | 0.314 | 0.379 | 0.135 | 0.376 | 0.478 | | |
| in100+laion100-VL-ttd | 0.529 | 0.438 | 0.289 | 0.299 | 0.107 | 0.283 | 0.536 | N/A | N/A |
| **Data Budget: 10x** | | | | | | | | | |
| **JANuS-int** | 0.908 | 0.863 | 0.678 | 0.731 | 0.35 | 0.656 | 0.722 | N/A | N/A |
| JANuS-int-384-res | 0.868 | 0.792 | 0.559 | 0.622 | 0.355 | 0.582 | 0.671 | N/A | N/A |
| JANUS-int-RN50x4 | 0.909 | 0.866 | 0.66 | 0.718 | 0.387 | 0.658 | 0.724 | N/A | N/A |
| JANuS-int-ViT-S-16 | 0.823 | 0.736 | 0.4 | 0.484 | 0.237 | 0.464 | 0.564 | N/A | N/A |
| JANuS-int-gt+swinlabels-1.1m | 0.908 | 0.863 | 0.678 | 0.731 | 0.349 | 0.655 | 0.721 | N/A | N/A |
| JANuS-int-gt+sbm-1.1m | 0.871 | 0.817 | 0.625 | 0.659 | 0.276 | 0.594 | 0.682 | N/A | N/A |
| JANuS-VL-gt+ttd-1.1m | 0.846 | 0.757 | 0.447 | 0.506 | 0.204 | 0.478 | 0.566 | N/A | N/A |
| JANuS-VL-ofa-1.1m | 0.67 | 0.587 | 0.392 | 0.453 | 0.147 | 0.395 | 0.589 | N/A | N/A |
| **Data Budget: 20x** | | | | | | | | | |
| **JANuS+yfcc-2.4m-int-cliplabels** | 0.927 | 0.877 | 0.7 | 0.78 | 0.449 | 0.702 | 0.757 | N/A | N/A |
| JANuS+yfcc-2.4m-int-cliplabels-ViT-S-16 | 0.878 | 0.829 | 0.508 | 0.596 | 0.372 | 0.576 | 0.656 | N/A | N/A |

Table 9: **Approach 1; Full results.** *Results for models trained from scratch using Approach 1 (JANuS).*

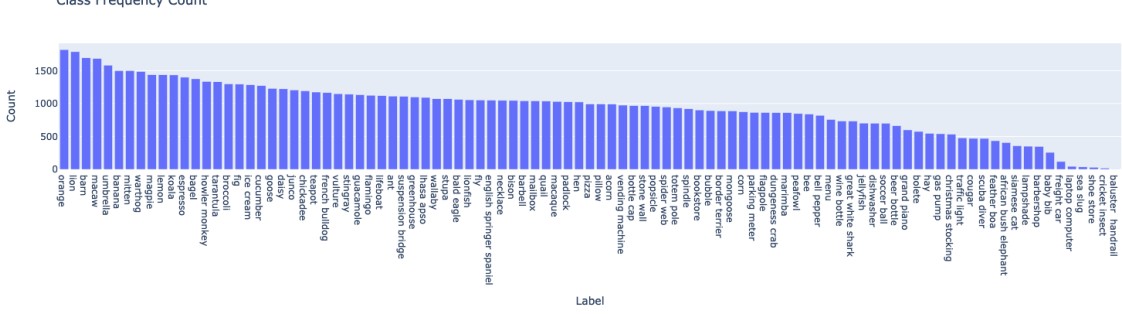

Figure 15: **Class frequency count using subset matching on ImageNet-100.**

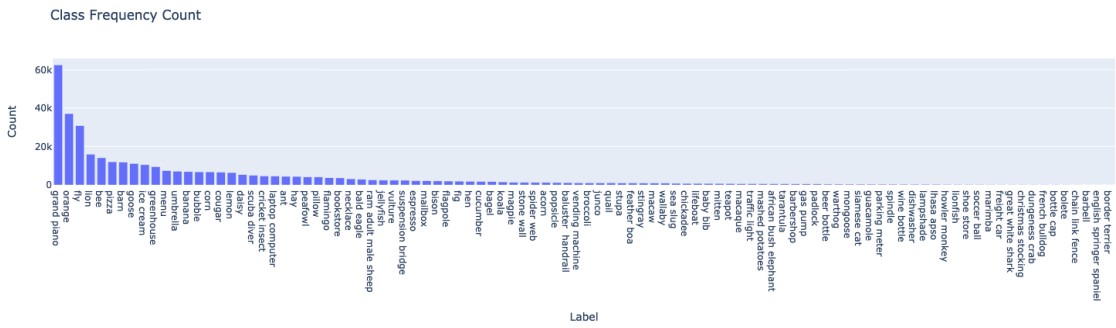

Figure 16: **Class frequency count using subset matching on YFCC-100.**

# J  Class Frequency Counts for IN100 subset matching distributions, openai labels, mc matching

The figures referenced below depict the distribution of noisy, web-scraped labels with respect to true labels for each of the constituent datasets in JANuS.

ImageNet-100: 15. YFCC-100: 16. LAION-100: 17.

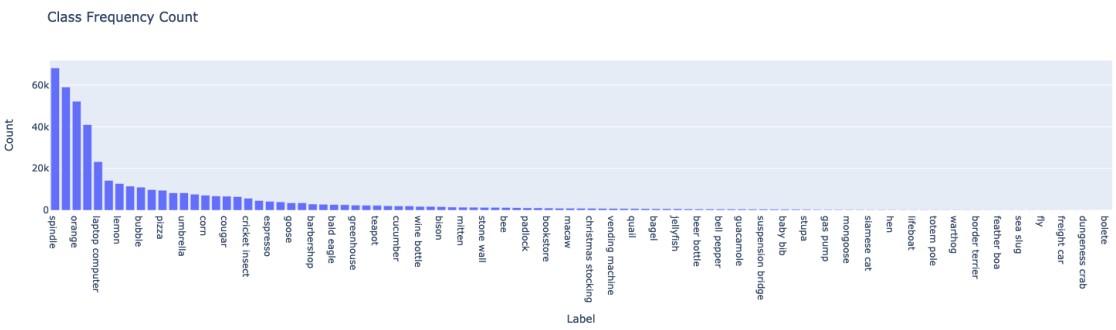

Figure 17: **Class frequency count using subset matching on LAION-100.**

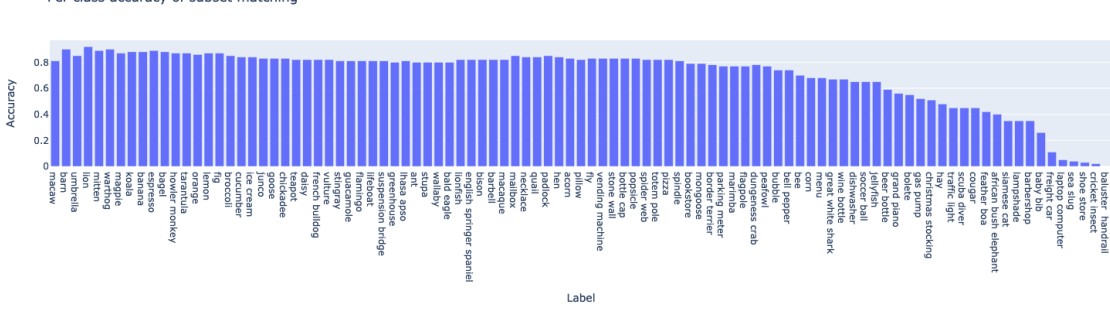

Figure 18: **Per class accuracy when using subset matching on IN-100.**

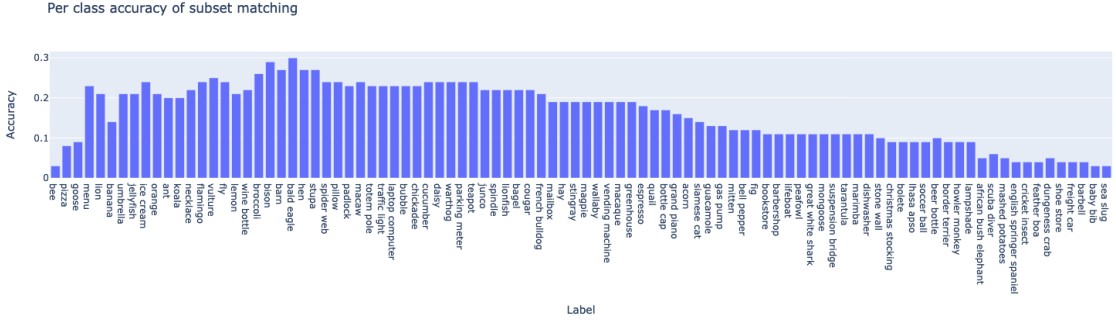

Figure 19: **Per class accuracy when using subset matching on OI-100.**

## K   Per Class Accuracy for Subset Matching, openai classnames, sc

We report the accuracy of the subset-matched labels, compared to ground-truth labels, for the following constituent datasets in JANuS:

ImageNet-100: 18. OpenImages-100: 19.

## L   JANuS Spreadsheet Column Explanations

JANuS contains many different kinds of metadata, and the meaning of some of the column labels used may not be immediately apparent to the reader.

We do not provide explanations for metadata columns which are explained in one of the original dataset descriptions; for those, we recommend referring to the original authors of the datasets. (Deng et al., 2009; Fang et al., 2022; Schuhmann et al., 2021a; Thomee et al., 2016; Kuznetsova et al., 2020)

**BLIPCaption** refers to captions generated by us using a BLIP captioning model. **BLIPTitle** captions are a combination of the BLIP caption and the title field of flickr captions. Li et al. (2022a)

**FlickrCaption** refers to captions sourced from flickr.

**annot_caption** refers to OpenImages captions that were authored by human image annotators. **prose_caption** combines BLIP and annotator captions, favoring the latter when available.

**clip_idx** are ImageNet labels chosen by a zero-shot CLIP ViT-L model from OpenAI.

**idx_** labels refer to labels generated using various subset-matching strategies.

**mc** is multiclass, **sc** is single class, **strict** is strict. **Ours, default, openai** refer to the three different sets of class labels we experimented with throughout this paper.

