# OpenReview forum: "Distributionally Robust Classification on a Data Budget"
_TMLR — Accepted by TMLR_

### Review · Reviewer_bBmd · 2023-06-13

**Summary Of Contributions:**

This paper presents a new study focused on identifying the primary design features that contribute to distributional robustness in contemporary deep learning practices. By exploring various dimensions of variation, including model size, architecture, data diversity, and training protocols, the authors aim to uncover the crucial factors that underpin robustness. To facilitate controlled scaling experiments, the paper introduces JANuS, a novel training dataset designed to elucidate the impact of different data scales on robustness in modern deep learning. JANuS is a compilation of four publicly available training datasets that have undergone major cleaning and unification processes. It features a single label and caption space, ensuring compatibility with the ImageNet-100 dataset and its shifted variants, which serve as the main basis for evaluations conducted in this study. At the end, the paper states some guidelines for practitioners seeking to enhance robustness.

**Audience:**

Yes

**Broader Impact Concerns:**

I don't see any clear negative societal impacts directly stemming from this work.

**Claims And Evidence:**

No

**Requested Changes:**

To recommend acceptance, I would require a substantial overhaul of Section 6 in the revised version of the manuscript. It is crucial that this section is divided into different subsections to improve clarity and organization. Each experiment should focus on studying a single factor of variation in isolation, while keeping all other variables fixed. While some degree of stratification, such as distinguishing between VL-loss and CE-loss in certain cases, may be acceptable, it should be properly justified. The key aspect is that each experiment should convey a clear and unique message to the reader.

To enhance the comprehensibility of the findings, it is paramount that every statement is supported by relevant tables or figures. The most significant and important findings should be presented directly in the main text, while any supplementary or less central information can be deferred to the appendix. It is essential to minimize the need for constant referrals to the appendix, ensuring that the main paper provides a comprehensive overview. Care should be taken to avoid missing details, and all relevant information should be included in the text. I expect to see evidence that each of the guidelines in Section 7 are properly supported by a clean experiment.

If these changes are not implemented to a degree that makes the presented findings reliable, I will regrettably recommend rejection of the manuscript.

**Strengths And Weaknesses:**

## Strengths

#### 1. **Topic of great interest**:
The paper addresses a highly relevant and actively explored research area. Understanding the key factors that contribute to robustness in modern deep learning practices is of great interest to the wide TMLR audience.

#### 2. **JANuS could be a useful dataset**:
The release of the JANuS dataset is a valuable contribution of this work. The dataset provides a standardized, relatively clean dataset with access to labels and captions for all samples. This will facilitate future studies aiming to analyze the differences between vision-language models and image-only models. I also agree with the authors that the coarser granularity of the label space of JANuS enables the development of high-performing models at a lower training cost, which could potentially democratize research in this domain.

## Weaknesses

#### 1. (Major) **Cumbersome presentation of results and lack of experimental details**:
The primary drawback of this paper lies in the cluttered and poorly motivated presentation of results in Section 6. The experimental settings for each "ablated" factor of variation are unclear, and the rationale behind the experiments is not adequately explained. The paper fails to deliver on its promise of conducting a controlled and unified benchmarking of distinct approaches, as the presented results mix multiple factors of variation without proper unity. The mix usage of models pre-trained on different sources, the inconsistent treatment of sample numbers, or free use of different architectures further exacerbates the lack of clarity of the section. Overall, the narrative is extremely challenging to follow, as the experiments are not properly described, making it difficult for readers to draw meaningful conclusions.

As an example, the paragraph discussing the **Model inference comparison** should focus on the effect of data scale. However, the results also delve into a comparison between VL-loss and CE-loss, along with an experiment involving different ways to filter examples and a comment about the impact of using various pre-training datasets (with missing details). While these experiments might not be inherently wrong, their mixed presentation and possibly inconsistent execution render them incomprehensible for readers seeking meaningful insights.

In general, the entire Section 6 suffers from similar issues and is ridden with missing experimental details. Many times in the text the authors comment on findings for which no supporting table or figure is provided. Most importantly, the whole section **E. Model Training Details** is missing from the appendix This is problematic, as the lack of a consistent experimental setup, along with the absence of supporting tables and figures in the missing further hinders the reader's understanding and compromises the validity of the findings.

#### 2. **Unclear conclusions**:
The difficulties posed by the mixed and convoluted experiments make it challenging to derive clear conclusions from this work. The guidelines presented in Section 7 lack a solid experimental foundation, as they are intertwined with numerous confounding factors. Moreover, while the JANuS dataset holds potential value, its utilization is unclear, as the paper fails to clearly indicate when it is used and when it is not, undermining its potential contribution to the readers.

#### 3. (Minor) **Formatting errors**:
Several formatting errors are present throughout the paper: The usage of `\citep` and `\citet` macros in LaTeX is frequently incorrect. Additionally, the decimal numbers in the tables inconsistently exhibit a leading zero, and the sudden transition in the appendix from a single-column format to a multi-column format is inconsistent and requires attention.

---

> ### Author Response · Authors · 2023-07-11
> **Reply to reviewer bBmd**
>
> We thank you for your detailed comments. We share your view that JANuS is a useful dataset for future research, and that the topic is of interest to the TMLR audience.
>
> We wish to profusely apologize for the unclear presentation of results in the previous draft, and the unclear support for the conclusions which derive from them. Upon reflection, we agreed with this assessment, and have taken the following steps to attempt to address this concern:
>
> * We have divided our results into two distinct experimental approaches. Approach 1 is a controlled study of models trained on the JANuS dataset. Approach 2 is a meta-analysis of over 650 models trained on a diverse range of pre-training datasets. We discuss the motivations for this dual approach in Section 5 of our revised draft, along with implementation details for each approach.
>
> * We have enumerated four foundational questions we intend to address with our analysis. We have listed our results for each question in Section 6. To simplify understanding we removed findings which do not clearly contribute to answering one of these questions.
>
> * We have divided our results in Section 6 to clearly indicate whether a result we reference was derived from Approach 1 or Approach 2, so as to segregate fully controlled experiments from large-scale meta-analysis results.
>
> * We have refined the complete list of our Approach 1 models by reformatting Table 8 in the appendix and adding new descriptions in Section H.
>
> * We have attached a previously missing supplement which lists all of the experimental results for Approach 2.
>
> * We have corrected the previous Section E (new Section F) in the appendix so that it clearly states the provenance of all models used in our Approach 2 (meta-analysis).
>
> We have also attempted to correct the minor formatting errors you reference in point 3.
>
> Overall, we hope that these responses address any questions you had regarding the paper.

---

> > ### Comment · Reviewer_bBmd · 2023-07-28
> > **Reply to authors**
> >
> > I thank the authors for having followed my recommendations and significantly improved the writing an exposition of the paper. The new structure is much better than the original one and has helped solve my main concerns. I agree with the other reviewers that the findings of this work are not very surprising, but I do not think this is a reason to reject the paper. In my opinion, the exposition of the paper could be further improved, but again I don’t see this as a reason for rejection anymore. Overall, I lean to accept this paper, as the JANuS is a nice contribution, and some of the experiments provide further corroboration of some intuitions in the literature.

---

### Review · Reviewer_FEFA · 2023-06-19

**Summary Of Contributions:**

This paper empirically investigates the factors affecting image classifiers' robustness against natural distribution shifts. By evaluating on various datasets, architectures, and training data sizes, the authors found that we can train robust learners in a domain where data is limited. It seems that the key to success is the large and fine-grained label sets.



**Audience:**

Yes

**Claims And Evidence:**

Yes

**Requested Changes:**

Please address the issues in the Weaknesses part.

**Strengths And Weaknesses:**

# Strengths
- The empirical studies are thorough, especially the ablation studies.
- The paper is clearly written.

# Weaknesses

- It seems that the large and fine-grained label sets contribute to the success of this paper. What is the underlying reason? Has the model witnessed enough variations?

- We know CLIP can work on noisy text supervision. Besides, CLIP is not targeted for image classification---it can serve as general feature extractors, as in diffusion models and mini-gpt4. So, is the comparison between your method on CLIP fair?

- What if we fine-tune CLIP to handle the problem you mentioned? I think CLIP can serve as a good warmup.

- Overall, the conclusion that large and fine-grained label sets yield better robustness is unsurprising. And honestly, we cannot regard 2.4M samples as a limited budget. So I question the significance of this paper.

# Minor
- what is the SWSL in table 1?

---

> ### Author Response · Authors · 2023-07-11
> **Reply to reviewer FEFA**
>
> Thank you for your thoughtful comments! We are very happy that you found our paper clearly written, and our empirical findings thorough!
>
> You are correct that we found both the size and the coarseness of the label set can strongly affect distributional robustness. The underlying reason that larger label sets are less robust is most likely because conventional softmax classification becomes less effective; see https://arxiv.org/abs/1812.05737 and https://arxiv.org/pdf/1905.10626.pdf for more on this point. While this effect was known to exist for extremely large label sets, our contribution was to show that it can be extremely impactful even for (relatively) small label sets like ImageNet-1k, the standard benchmark. To your other point, coarser label sets are easier to disambiguate using fewer samples; it's harder to classify a dog as one of 100 breeds based on an image than it is to tell whether the image is of a dog or a carrot. Therefore, as you say, we postulate the model can witness enough variations to become robust faster when training on a small, coarse label set. To address this in more detail, we have added a discussion of this topic to Section D.2 in the appendix.
>
> Regarding your question about CLIP being a general feature extractor: while it is true that CLIP embeddings are generally usable, the original CLIP paper presented extensive empirical results on image classification, including accuracy and robustness on ImageNet. Furthermore, several downstream uses of CLIP embeddings do depend on accurate and robust classifications. Finally, measuring robustness of zero-shot CLIP-based classifiers is standard in this literature (see for example, Fang et al., “Data Determines Distributional Robustness”). Overall, we feel that evaluating on ImageNet shifts are indeed an important indicator of CLIP’s true robustness.
>
> Your suggestion about fine-tuning CLIP is very reasonable; we discuss some of the strengths and limitations of fine-tuning in Section 3 of the main paper, as well as Appendix A of our updated draft. While large pretrained vision models fine-tuned on a new task tend to outperform models trained from scratch with respect to base accuracy, others (including the original CLIP paper of Radford et al.) have shown that fine-tuning rapidly erodes distributional robustness in zero-shot models such as CLIP. In our appendix, we also show that interpolation methods such as Wise-FT and LiT-tuning can improve distributional robustness compared to conventional fine-tuning, but still fall short of zero-shot performance. Finally, in Table 4, we observe that both ViTs and ResNets fail to preserve robustness when fine-tuned after large-scale pretraining.
>
> Regarding your observation that 2.4 million samples might not be considered a limited data budget: this is a matter of terminology, but we argue that it is limited compared to the scale of CLIP pre-training (400 million samples). We also show that even with half as many samples, a CE-loss model is able to come fairly close to CLIP in terms of robustness. We also release the dataset itself, which we believe is of general interest to the TMLR audience as a testbed for future experiments.
>
> Overall, we hope that these responses address any questions you had regarding the paper.

---

### Review · Reviewer_DK2J · 2023-06-28

**Summary Of Contributions:**

The paper addresses the challenge of training robust deep learning models under distribution shifts, even with limited data. The authors introduce a new dataset called JANuS, which includes images, labels, and captions, and use it to investigate factors contributing to robustness in image classification.
The authors demonstrate that it is possible to train highly robust and accurate models using conventional cross-entropy loss, even with limited data and model sizes. They conduct a comprehensive analysis  and show that modest scaling of models and data can lead to robust models on large and fine-grained label sets.
The paper's contributions include showing the feasibility of training robust models with limited data and model sizes, introducing the JANuS dataset, conducting a large-scale analysis of image classification models, providing heuristics for improving distributional robustness with limited data, and releasing code, dataset, and results for reproducibility.
Overall, the paper provides valuable insights into training robust deep learning models under limited data conditions and offers resources for future research in distributional robustness.

**Audience:**

Yes

**Broader Impact Concerns:**

Not applicable.

**Claims And Evidence:**

Yes

**Requested Changes:**

It would be great if the authors of the paper can describe further about the proposed JANuS dataset, and provide some metrics for us to further understand the dataset. I see that additional descriptions are available in the appendix, but it might be beneficial to include some of the details in the main text, as I think the dataset is a major contribution of the paper.

**Strengths And Weaknesses:**

Strengths:
- All in all, the paper is well written and easy to follow.
- The introduction of JANuS, provides a unique resource for studying robustness in image classification. Such a dataset can significantly facilitate future research.
- The paper demonstrates the possibility of training highly robust models using conventional cross-entropy loss, even when data and model sizes are limited.
- The authors of the paper provide comprehensive experimental analysis. This large-scale evaluation enhances the reliability and generalizability of the findings.
- The paper outlines practical heuristics for improving distributional robustness when faced with limited data budgets.

Weaknesses:
- The paper might have limited algorithmic novelty.
- Since the JANuS dataset is derived from several existing datasets based on automated selection criteria like subset matching as described in the paper. This can cause issues like dataset bias and overfitting.

---

> ### Author Response · Authors · 2023-07-11
> **Reply to reviewer DK2J**
>
> Thank you for your thoughtful comments! We are very happy that you found our paper well written and easy to follow!
>
> We agree with your evaluation that JANuS can serve as an important resource for future research on distributional robustness, and that combining large-scale and small-scale experiments enhances both the reliability and generalizability of results derived from JANuS.
>
> Please note that we do not claim any algorithmic novelty in the manuscript. We feel that this should not be held against the paper, since algorithmic novelty is not a criterion listed for acceptance in the TMLR guidelines. Further, we believe that the insights provided by our careful evaluation studies (both on controlled datasets as well as meta-analyses) are interesting in their own right, and therefore merit publication.
>
> We agree that any method of dataset filtration, including our proposed subset matching on noisy web-scraped captions, can unintentionally introduce dataset bias. In Section 4 of our updated draft, we describe experimental results on JANuS which confirm that the performance of JANuS-trained models on ImageNet-100 validation is comparable to models trained on ImageNet-1000 and validated on ImageNet-100. This is suggestive of the fact that performance on JANuS is a relatively unbiased, faithful proxy for performance on ImageNet.
>
> With any small dataset, including JANuS, overfitting is indeed a concern; this is a natural consequence of our focus on limited data regimes. Since we measure distributional robustness as well as base accuracy, we can to some extent estimate the degree of overfitting which takes place on JANuS – this idea motivated many of our experiments.
>
> We appreciate your suggestion that we describe further about the proposed JANuS dataset.  In Section 4 of the updated draft, we provide several metrics to further understand the dataset, and make available additional details about our curation method. Our updated Table 7 in the Appendix provides metrics on the quantity of samples available and the search terms used to source images.
>
> Overall, we hope that these responses address any questions you had regarding the paper.

---

### Author Response · Authors · 2023-07-11
**Rebuttal Comment**

We wish to thank all of the reviewers for their insightful comments and for recognizing the strengths of our contributions to the literature; a combination of carefully controlled comparisons on a novel dataset, JANuS, with a large-scale meta-analysis of distributional robustness, allowing us to make empirically validated suggestions for how to achieve distributional robustness on a data budget.

We also appreciate their clarifying questions and suggestions, and have responded to those questions and suggestions, point by point, in the individual replies below. At a high level, our updated manuscript reflects the following changes:

* We completely revamped Sections 4, 5, and 6 of the previous manuscript as a response to Reviewer bBmd’s concerns regarding clarity of presentation of our results. We carefully separate our designed experiments (and corresponding results) into two categories: controlled experiments on JANuS, and large-scale meta-analyses; infer trends based on the results of these experiments; and connect them to specific findings.

* We added findings from newer experiments on fine-tuning, as a response to a comment from Reviewer FEFA; in particular, we note that both CLIP fine-tuning (Appendix A) and fine-tuning ViTs and ResNets fails to preserve robustness after large-scale pretraining.

* We improved presentation overall (and in particular, that of the Appendix); the paper is now hopefully more self-contained, while still maintaining sufficient descriptiveness.

---

### Decision · Action_Editors · 2023-08-01

**Recommendation:** Accept as is

**Comment:**

Summarising reviewers final comments, post revision reviewers' concerns regarding the lack of evidence for the results have been significantly reduced. The paper reads better, although it still could be improved.



**Audience:**

Yes

**Claims And Evidence:**

After reviews and back and forth discussion with the authors, the revision seems to provide the necessary details for claims and evidence